# The natural history of symptomatic COVID-19 during the first wave in Catalonia

Edward Burn [1,2], Cristian Tebé[3,4], Sergio Fernandez-Bertolin[1], Maria Aragon [1], Martina Recalde[1,5], Elena Roel [1], Albert Prats-Uribe [2], Daniel Prieto-Alhambra [2✉] & Talita Duarte-Salles[1]

The natural history of coronavirus disease 2019 (COVID-19) has yet to be fully described. Here, we use patient-level data from the Information System for Research in Primary Care (SIDIAP) to summarise COVID-19 outcomes in Catalonia, Spain. We included 5,586,521 individuals from the general population. Of these, 102,002 had an outpatient diagnosis of COVID-19, 16,901 were hospitalised with COVID-19, and 5273 died after either being diagnosed or hospitalised with COVID-19 between 1st March and 6th May 2020. Older age, being male, and having comorbidities were all generally associated with worse outcomes. These findings demonstrate the continued need to protect those at high risk of poor outcomes, particularly older people, from COVID-19 and provide appropriate care for those who develop symptomatic disease. While risks of hospitalisation and death were lower for younger populations, there is a need to limit their role in community transmission.

[1] Fundació Institut Universitari per a la recerca a l'Atenció Primària de Salut Jordi Gol i Gurina (IDIAPJGol), Barcelona, Spain. [2] Centre for Statistics in Medicine (CSM), Nuffield Department of Orthopaedics, Rheumatology and Musculoskeletal Sciences (NDORMS), University of Oxford, Oxford, UK. [3] Biostatistics Unit at Bellvitge Biomedical Research Institute (IDIBELL), L'Hospitalet, Barcelona, Spain. [4] Universitat de Barcelona, Barcelona, Spain. [5] Universitat Autònoma de Barcelona, Bellaterra, Spain. ✉email: daniel.prietoalhambra@ndorms.ox.ac.uk

The natural history of coronavirus disease 2019 (COVID-19) has yet to be well-established. While a wide range of studies have described outcomes among patients hospitalised with COVID-19[1,2], these individuals represent only a fraction of those who develop symptomatic disease. Similarly, while outcomes for tested populations have also revealed important insights[3,4], COVID-19 testing has often been prioritised on the basis of individuals' symptoms or perceived risk of outcomes, making inferences about specific risk factors for progression among these populations difficult[5].

A full description of the natural history of COVID-19 from symptomatic to severe disease is needed. Such a description requires comprehensive patient-level data that captures incident, symptomatic cases from a representative population, with subsequent longitudinal follow-up, and where outcomes such as hospitalisations and mortality, both inside and outside of the hospital setting, can be identified. Moreover, assessing the impact of chronic health conditions on the course of the disease requires comprehensive data on patients' medical histories. Linked real world data from countries with universal taxpayer-funded primary care-based health systems where general practitioners are the first point of contact for care, and where this role has been maintained in the response to COVID-19, provide a unique opportunity for this purpose.

In Spain, one of the European countries worst hit by the COVID-19 pandemic, primary care has continued to play an important role in the response to the disease. In Catalonia, an autonomous region of Spain with a devolved health system, more than 120,000 outpatient cases of COVID-19 were diagnosed between 15th March and 24th April 2020[6]. Meanwhile, a nationwide seroprevalence study conducted between 27th April to 11th May, 2020 found the prevalence of IgG antibodies to be around 5% in Spain, and 7% in Catalonia[7]. Consequently, despite the burden of disease already experienced, there is a need to better understand the features of COVID-19 so as to inform the continued regional, national, and global responses to it.

In this study, our first aim was to summarise COVID-19 related outcomes in Catalonia as experienced during the first wave of the pandemic. Subsequently, we aimed to describe the associations between age, sex, and comorbidities and risks of COVID-19 diagnosis, hospitalisation with COVID-19, and a COVID-19-related death during the first wave of COVID-19 in Catalonia. It should be stressed that these latter research questions are inferential, assessing the existence of relationships but not the underlying mechanisms or reasons for them[8].

## Results

**Study participants and observed outcomes.** A total of 5,586,521 participants were included in the study (Fig. 1). Based on the study eligibility criteria described above, 169,575 individuals were excluded due to a lack of a year of prior history, one for having a COVID-19 positive test prior, 307 for having a prior COVID-19 diagnosis, and three for having a COVID-19 hospitalisation prior to the 1st March 2020. A further 1207 individuals who were hospitalised on the 1st March 2020 and 40,999 who were care home residents were also excluded (a flow chart is provided in Supplementary Fig. 1). The characteristics of the final study population are summarised in Table 1. A histogram of the age distribution of the study population can be seen in Fig. 2, with a summary of comorbidities shown in Fig. 3.

Out of the included study population, 102,002 went on to have an outpatient diagnosis of COVID-19 (67-day cumulative incidence: 1.83%). Of those diagnosed in outpatient settings, 2581 died with COVID-19 without being hospitalised (45-day cumulative incidence: 3.01%). In total, 16,901 individuals had a hospitalisation with COVID-19, with 8860 of them having previously had an outpatient diagnosis (45-day cumulative incidence: 9.06%). Of those hospitalised, 2692 had died with COVID-19 by the end of follow-up (45-day cumulative incidence: 19.16%), see Fig. 1 and Table 2. An extensive summary of the observed data can be seen alongside the study code at https://github.com/SIDIAP/MultiStateCovid-19.

**The association between age and risk of COVID-19 diagnosis, hospitalisation, and death.** Age profiles varied by transition, see Table 1. While the median age of the study population as a whole was 43 (interquartile range: 25–60), those diagnosed with

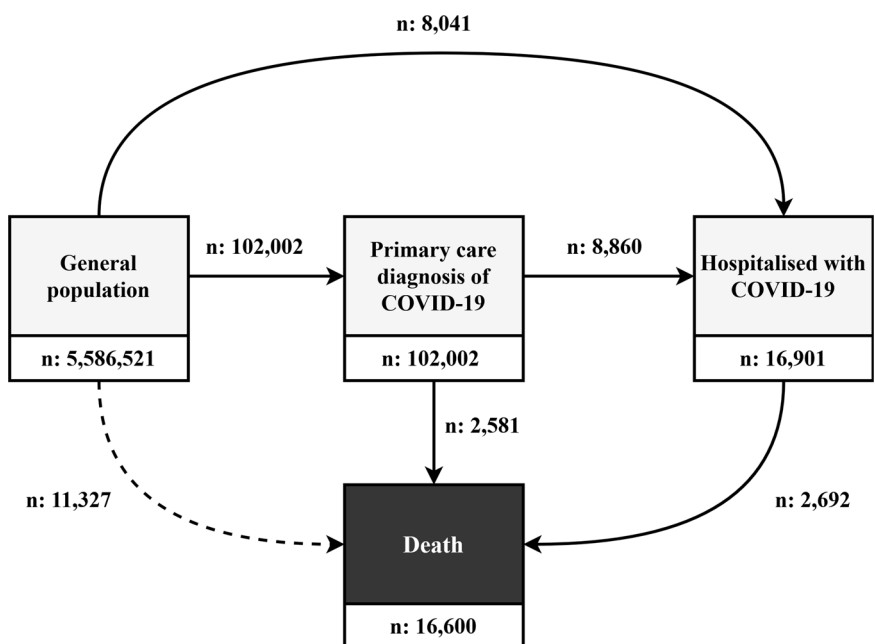

**Fig. 1 Multi-state model of COVID-19.** The entire study population began in the general population state as of the 1st March 2020, with progression through the model possible up to 6th May 2020.

**Table 1 Patient characteristics of the study population and by multistate transition.**

| | General population | From general population | | | From diagnosed with COVID-19 | | From hospitalised with COVID-19 |
| | | To diagnosed with COVID-19 | To hospitalised with COVID-19 | To death | To hospitalised with COVID-19 | To death | To death |
|---|---|---|---|---|---|---|---|
| *N* | 5,586,521 | 102,002 | 8041 | 11,327 | 8860 | 2581 | 2692 |
| Days since index date (median [IQR]) | | | | | 24 [19.0, 32.0] | 33 [27.0, 40.0] | 26 [21.0, 35.0] |
| Age (median [IQR]) | 43 [25.0, 60.0] | 46 [35.0, 58.0] | 71 [58.0, 80.0] | 85 [76.0, 90.0] | 60 [49.0, 73.0] | 86 [80.0, 91.0] | 80 [73.0, 86.0] |
| *Age gr (%)* | | | | | | | |
| Under 18 | 966,680 (17.3) | 4536 (4.4) | 34 (0.4) | 9 (0.1) | 40 (0.5) | <5 | <5 |
| 18–39 | 1,437,297 (25.7) | 30,530 (29.9) | 426 (5.3) | 80 (0.7) | 850 (9.6) | 5 (0.2) | 11 (0.4) |
| 40–59 | 1,785,852 (32.0) | 44,493 (43.6) | 1776 (22.1) | 657 (5.8) | 3441 (38.8) | 63 (2.4) | 142 (5.3) |
| 60–69 | 615,538 (11.0) | 10,305 (10.1) | 1558 (19.4) | 1026 (9.1) | 1673 (18.9) | 131 (5.1) | 292 (10.8) |
| 70–79 | 468,662 (8.4) | 6120 (6.0) | 2162 (26.9) | 1967 (17.4) | 1602 (18.1) | 406 (15.7) | 822 (30.5) |
| 80 or older | 312,492 (5.6) | 6018 (5.9) | 2085 (25.9) | 7588 (67.0) | 1254 (14.2) | 1976 (76.6) | 1424 (52.9) |
| Sex = female (%) | 2,831,062 (50.7) | 59,006 (57.8) | 3439 (42.8) | 5672 (50.1) | 3989 (45.0) | 1467 (56.8) | 1075 (39.9) |
| *Region (%)* | | | | | | | |
| Barcelona | 4,169,520 (74.6) | 82,702 (81.1) | 7201 (89.6) | 8694 (76.8) | 7604 (85.8) | 2164 (83.8) | 2393 (88.9) |
| Girona | 557,006 (10.0) | 9929 (9.7) | 432 (5.4) | 942 (8.3) | 657 (7.4) | 217 (8.4) | 158 (5.9) |
| Lleida | 370,570 (6.6) | 5621 (5.5) | 211 (2.6) | 728 (6.4) | 391 (4.4) | 108 (4.2) | 94 (3.5) |
| Tarragona | 489,378 (8.8) | 3748 (3.7) | 196 (2.4) | 958 (8.5) | 207 (2.3) | 92 (3.6) | 45 (1.7) |
| Missing | 47 (0.0) | <5 | <5 | 5 (0.0) | <5 | <5 | <5 |
| *Charlson comorbidity index (%)* | | | | | | | |
| 0 | 4,558,137 (81.6) | 80,214 (78.6) | 3627 (45.1) | 1613 (14.2) | 5385 (60.8) | 365 (14.1) | 583 (21.7) |
| 1 | 400,005 (7.2) | 8996 (8.8) | 1036 (12.9) | 1607 (14.2) | 1031 (11.6) | 519 (20.1) | 427 (15.9) |
| 2 | 357,892 (6.4) | 6846 (6.7) | 1409 (17.5) | 2304 (20.3) | 1164 (13.1) | 460 (17.8) | 559 (20.8) |
| 3+ | 270,487 (4.8) | 5946 (5.8) | 1969 (24.5) | 5803 (51.2) | 1280 (14.4) | 1237 (47.9) | 1123 (41.7) |
| Autoimmune condition (%) | 273,496 (4.9) | 6251 (6.1) | 806 (10.0) | 1285 (11.3) | 727 (8.2) | 275 (10.7) | 326 (12.1) |
| Chronic kidney disease (%) | 202,102 (3.6) | 3934 (3.9) | 1408 (17.5) | 3710 (32.8) | 892 (10.1) | 887 (34.4) | 819 (30.4) |
| COPD (%) | 119,950 (2.1) | 2359 (2.3) | 740 (9.2) | 1457 (12.9) | 463 (5.2) | 261 (10.1) | 363 (13.5) |
| Dementia (%) | 42,579 (0.8) | 1933 (1.9) | 429 (5.3) | 2701 (23.8) | 294 (3.3) | 990 (38.4) | 376 (14.0) |
| Heart disease (%) | 533,261 (9.5) | 10,732 (10.5) | 2611 (32.5) | 5686 (50.2) | 1903 (21.5) | 1175 (45.5) | 1324 (49.2) |
| Hyperlipidemia (%) | 515,045 (9.2) | 10,820 (10.6) | 1498 (18.6) | 1580 (13.9) | 1507 (17.0) | 381 (14.8) | 508 (18.9) |
| Hypertension (%) | 689,436 (12.3) | 13,747 (13.5) | 2405 (29.9) | 3834 (33.8) | 2206 (24.9) | 896 (34.7) | 932 (34.6) |
| Malignant neoplasm (%) | 291,535 (5.2) | 5595 (5.5) | 1327 (16.5) | 3751 (33.1) | 991 (11.2) | 588 (22.8) | 670 (24.9) |
| Obesity (%) | 928,163 (16.6) | 20,761 (20.4) | 3216 (40.0) | 3184 (28.1) | 3103 (35.0) | 695 (26.9) | 1088 (40.4) |
| Type 2 diabetes (%) | 317,505 (5.7) | 6000 (5.9) | 1644 (20.4) | 2612 (23.1) | 1284 (14.5) | 590 (22.9) | 710 (26.4) |

Counts of less than 5 have been obscured to protect patient privacy.
*COPD* chronic obstructive pulmonary disease, *IQR* interquartile range.

COVID-19 were 46 (35–58), and those hospitalised without an outpatient diagnosis were 71 (58–80). Individuals hospitalised after a diagnosis of COVID-19 were on average 60 (49–73) years old, and those who died after being diagnosed with COVID-19 (but who were not admitted to hospital beforehand) had an median age of 85 (76–90). Individuals who died after being hospitalised had a median age of 80 (73–86).

Estimated hazard ratios for age are shown in Fig. 4 and summarised in full detail in Supplementary Tables 1 to 19. A non-linear relationship can be seen for outpatient diagnosis with COVID-19, with a peak in risks among those aged around 45. Relative to a reference age of 65, estimated hazard ratios were 0.70 (95% confidence interval: 0.69–0.72) for a 20-year-old, 1.57 (1.55–1.60) for a 45-year-old in the overall model, with age as the sole explanatory factor. For those at oldest ages, relative hazard ratios differed in March as compared to April (ie there was non-proportionality in hazards, which can also be seen in the log-log plot shown in Supplementary Fig. 2). In March, oldest age was associated with a lower risk of diagnosis relative to a 65-year-old (with a hazard ratio of 0.65 [0.62–0.68] for a 90-year-old relative

to a 65-year-old), but in April risks were highest for oldest ages (with a hazard ratio of 1.68 [1.61–1.76] for a 90-year-old relative to a 65-year-old).

Risk of hospitalisation with COVID-19 after an outpatient diagnosis of COVID-19 peaked from around age 75, with estimated hazard ratios of 0.24 (0.23–0.26) for a 45-year-old, 1.20 (1.17–1.23) for a 70-year-old, and 1.22 (1.14–1.31) for a 90-year-old, all relative to a reference age of 65 years old.

Older age was associated with an increased risk of hospitalisation with COVID-19 without a prior diagnosis, death after being hospitalised with COVID-19, and, in particular, death after being diagnosed but without being hospitalised. An age of 90 years old was associated with hazard ratios, relative to 65 years old, of 2.91 (2.75–3.08), 8.28 (7.34–9.34), and 26.00 (23.62–28.62) for each of these, respectively. For an age of 20, these relative hazard ratios were all estimated to be less than 0.1. Models estimated separately by region gave broadly comparable results, see Fig. 4. Associations between age and diagnosis with COVID-19 varied by sex, with the peak in risk at middle age more pronounced among women, while increased risks at oldest ages were seemingly driven by

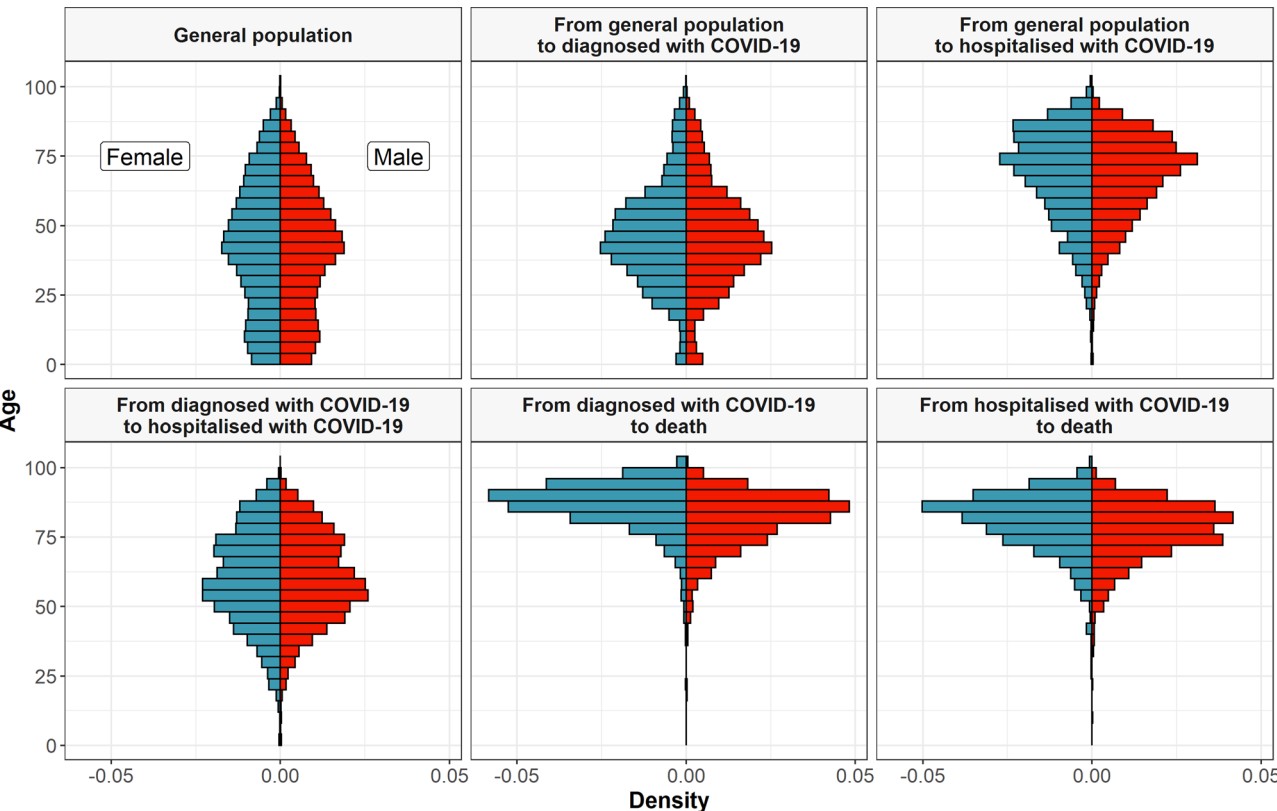

**Fig. 2 Histogram of age, by sex, for the study population and by transition in the multistate model.** The ages, split by sex, of the study population as a whole (general population) and of those individuals making transition from general population to diagnosed with COVID-19 and hospitalised with COVID-19, from diagnosed with COVID-19 to hospitalised with COVID-19 and death, and from hospitalised with COVID-19 to death.

males. Adjustment for comorbidities generally attenuated the associations for older ages to some degree, but not in their entirety (see Supplementary Tables).

**The association between sex and risk of COVID-19 diagnosis, hospitalisation, and death.** Compared to the study population as a whole, which was 51% female, a greater proportion of females were diagnosed with COVID-19 (58%), but more males (57%) transitioned directly to being hospitalised with COVID-19 without a previous outpatient diagnosis. Of those diagnosed, more males (55%) were subsequently hospitalised, but more females (57%) died without having been hospitalised. Of those hospitalised, 60% of those who died were male, see Table 1.

After adjustment for age, male sex was associated with a reduced risk of outpatient COVID-19 diagnosis (hazard ratio: 0.75 [95% CI: 0.74–0.75]). Conversely, male sex was associated with an increased risk of all other transitions (Fig. 5). Age-adjusted hazard ratios for males were 1.64 (1.57–1.72) for hospitalisation with COVID-19 without outpatient diagnosis, 1.77 (1.70–1.85) for hospitalisation after outpatient diagnosis of COVID-19, 1.33 (1.23–1.44) for death after outpatient diagnosis of COVID-19, and 1.31 (1.21–1.42) for death after hospitalisation with COVID-19. The overall increased risk of diagnosis with COVID-19 for females appeared to be driven by younger women, with an age-adjusted hazard ratio of 0.72 (0.71–0.73) for males, relative to females, among those 70 years old or younger, while the hazard ratio for those over 70 years old was 1.11 (1.07–1.15). The increased risk of death following diagnosis with COVID-19 appeared more pronounced among men younger than 70, while the increased risk of death following hospitalisation with COVID-19 for men appeared to be more pronounced in March (see Fig. 6 and Supplementary Table 17). Estimates were broadly consistent across by region, and further adjustment for

comorbidities made relatively little difference to these estimates (see Supplementary Tables 16 to 18).

**The association between comorbidities and risk of outpatient COVID-19 diagnosis, hospitalisation, and death.** Comorbidity profiles varied across different stages of the natural history of COVID-19, see Table 1 and Fig. 2. The highest proportion of otherwise healthy participants (Charlson index score of 0) was seen among those diagnosed with COVID-19 in outpatient settings (79%), and lowest amongst those who died after an outpatient diagnosis without hospital admission (14%). While, for example, the prevalence of chronic kidney disease, COPD, obesity, and type 2 diabetes were 4%, 2%, 17%, and 6%, respectively, among the study population as a whole, the prevalence of these conditions was 30, 14, 40, and 26% among those who died after being hospitalised with COVID-19. Of those individuals that died after diagnosis of COVID-19 without having being hospitalised, 38% had dementia (see Table 2).

After adjustment for age and sex, dementia had the strongest association with risk of outpatient diagnosis with COVID-19 (hazard ratio: 3.09 [95% CI: 2.94–3.24]), see Fig. 6. Aside from malignant neoplasm, all other comorbidities were associated with increased risks of outpatient diagnosis, with estimated hazard ratios between 1.08 (1.05–1.11) for type 2 diabetes to 1.23 (1.21–1.25) for obesity. Similarly, all conditions were associated with an increased risk of hospital admission for COVID-19 without a previous outpatient diagnosis, with obesity (hazard ratio: 1.78 [1.70–1.87]) and dementia (1.86 [1.68–2.06]) associated with the greatest excess risks.

Dementia was associated with a reduced risk of hospitalisation after diagnosis, but an increased risk of death without hospitalisation, with age and sex adjusted hazard ratios of 0.66

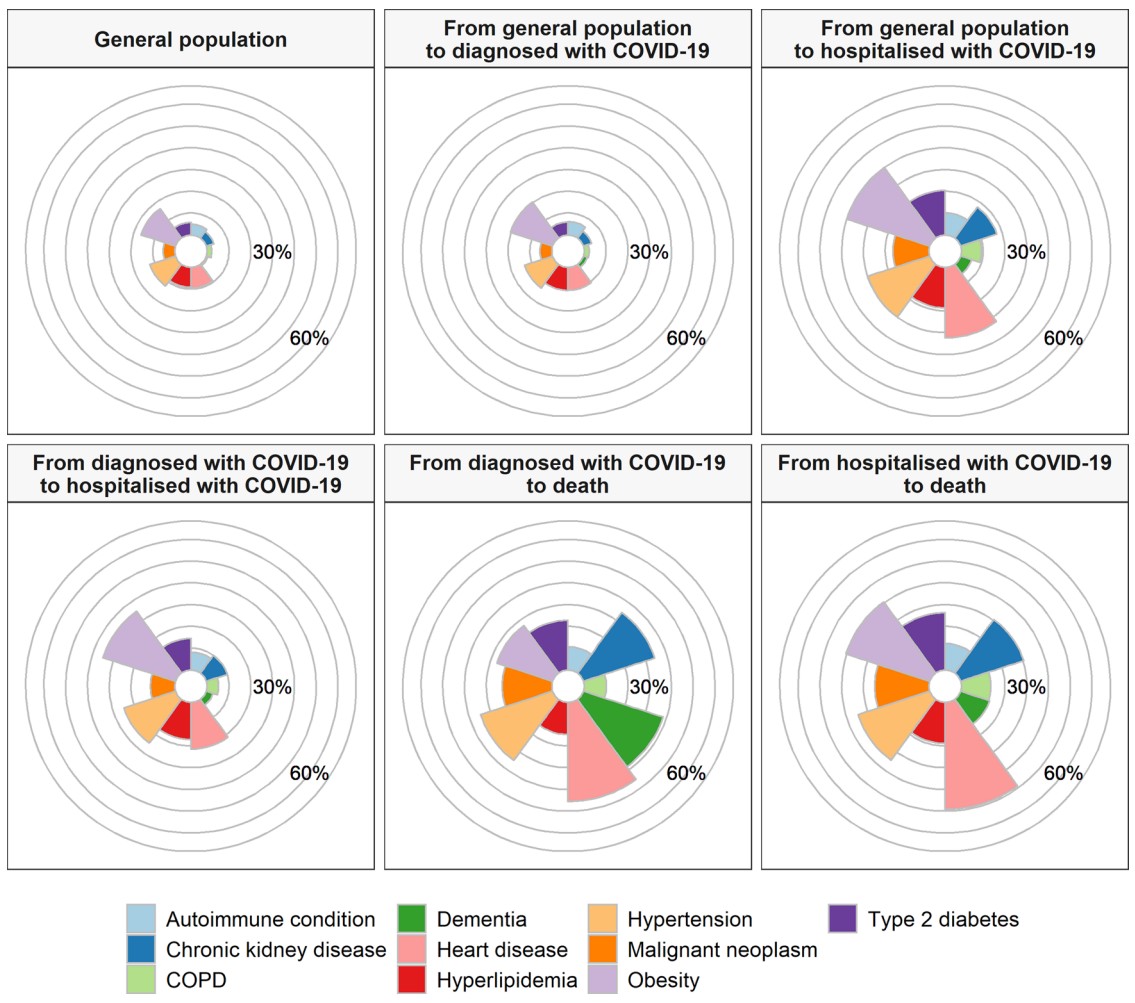

**Fig. 3 Prevalence of comorbidities among the general population and by transition in the multistate model.** The proportion of the study population as a whole (general population) with a comorbidity of interest, and of those individuals making transition from general population to diagnosed with COVID-19 and hospitalised with COVID-19, from diagnosed with COVID-19 to hospitalised with COVID-19 and death, and from hospitalised with COVID-19 to death. COPD chronic obstructive pulmonary disease.

(0.58–0.75) and 2.84 (2.60–3.09), respectively. Aside from COPD where no association was seen, all other conditions were associated with an increased risk of hospitalisation after outpatient diagnosis, ranging from a hazard ratio of 1.05 (0.98–1.13) for malignant neoplasm to 1.57 (1.50–1.64) for obesity.

While fewer conditions appeared associated with an increased risk of death following outpatient diagnosis (without hospitalisation), malignant neoplasm had a hazard ratio of 1.13 (1.03–1.24), chronic kidney disease had one of 1.24 (1.14–1.35), and type 2 diabetes one of 1.42 (1.29–1.56). Among those hospitalised, while little difference in outcomes was seen for those with hyperlipidaemia and hypertension after adjusting for age and sex, other conditions were associated with increased risks with hazard ratios ranging from 1.09 (1.01–1.18) for obesity to 1.98 (1.77–2.22) for dementia.

The association between the Charlson comorbidity index and transitions was most clearly seen for the risks of hospitalisation without a previous diagnosis and of death following an outpatient diagnosis with COVID-19 without hospitalisation. A score of three or more was, relative to a score of zero, associated with age and sex adjusted hazard ratios of 2.33 (2.18–2.48) and 2.59 (2.28–2.94), respectively.

Results for analyses stratified by age and sex are also shown in Fig. 6. It can be seen that various comorbidities were associated

with an even greater increase in risk for younger, and particularly female, study participants for a number of transitions. Obesity, for example, was associated with hazard ratios of 2.69 (2.43–2.97) and 2.11 (1.95–2.28) for hospitalisation from general population and hospitalisation after a diagnosis with COVID-19 among females, 70 years old or younger. While most associations between comorbidities and transitions were consistent across calendar time, dementia was associated with a greater increase in risk during April than in March. Dementia was associated with age and sex adjusted hazard ratios of 2.07 (1.91–2.25) and 1.33 (1.15–1.54) for transitions from general population to diagnosis with COVID-19 and hospitalisation with COVID-19 in March. These increased to 4.26 (4.00–4.54) and 2.93 (2.52–3.40) in April, respectively (see Supplementary Table 20 and Supplementary Fig. 3). Associations between comorbidities and transitions were seen to be broadly consistent across regions (see Supplementary Table 21 and Supplementary Fig. 4).

## Discussion

In this large cohort study, primary care data from 5,586,521 individuals with linked COVID-19 testing, hospitalisation, and mortality data has allowed for a detailed characterisation of the natural history of symptomatic COVID-19 as seen during the

**Table 2 Time at risk and observed outcomes by state populations.**

| | From general population | | | | From diagnosed with COVID-19 | | | From hospitalised with COVID-19 | |
| --- | --- | --- | --- | --- | --- | --- | --- | --- | --- |
| | Follow-up in days Median (min, IQR, max) | To diagnosis with COVID-19 Events (cumulative incidence at 67 days) | To hospitalised with COVID-19 Events (cumulative incidence at 67 days) | To death Events (cumulative incidence at 67 days) | Follow-up in days Median (min, IQR, max) | To hospitalised with COVID-19 Events (cumulative incidence at 45 days) | To death Events (cumulative incidence at 45 days) | Follow-up in days Median (min, IQR, max) | To death Events (cumulative incidence at 45 days) |
| All | 67 (1, 67-67, 67) | 102,002 (1.83%) | 8041 (0.14%) | 11,327 (0.20%) | 35 (0, 19-44, 66) | 8860 (9.06%) | 2581 (3.01%) | 36 (0, 25-43, 65) | 2692 (19.16%) |
| Age Under 18 | 67 (1, 67-67, 67) | 4536 (0.47%) | 34 (0.00035%) | 9 (0.00093%) | 29 (0, 16-40, 65) | 40 (0.96%) | <5 | 28 (0, 19-41.75, 55) | <5 |
| Age 18-39 | 67 (1, 67-67, 67) | 30,530 (2.13%) | 426 (0.03%) | 80 (0.0056%) | 37 (0, 22 to 44, 66) | 850 (2.91%) | 5 (0.03%) | 36 (0, 26.75-43, 65) | 11 (1.15%) |
| Age 40-59 | 67 (1, 67-67, 67) | 44,493 (2.49%) | 1776 (0.10%) | 657 (0.037%) | 37 (0, 22-44, 66) | 3441 (8.05%) | 63 (0.18%) | 37 (0, 29-43, 65) | 142 (3.54%) |
| Age 60-69 | 67 (1, 67-67, 67) | 10,305 (1.67%) | 1558 (0.25%) | 1026 (0.17%) | 34 (0, 14-44, 66) | 1673 (16.82%) | 131 (1.51%) | 38 (0, 30-44, 64) | 292 (10.80%) |
| Age 70-79 | 67 (1, 67-67, 67) | 6120 (1.31%) | 2162 (0.46%) | 1967 (0.42%) | 24 (0, 7-40, 65) | 1602 (27.21%) | 406 (8.07%) | 37 (0, 26-43, 65) | 822 (24.97%) |
| Age 80 or older | 67 (1, 67-67, 67) | 6018 (1.93%) | 2085 (0.67%) | 7588 (2.43%) | 16 (0, 7-31, 62) | 1254 (21.72%) | 1976 (40.84%) | 27 (0, 14-39.5, 65) | 1424 (51.31%) |
| Sex Male | 67 (1, 67-67, 67) | 42,996 (1.56%) | 4602 (0.17%) | 5655 (0.21%) | 35 (0, 16-44, 66) | 4871 (11.80%) | 1114 (3.13%) | 37 (0, 26-43, 65) | 1617 (20.46%) |
| Sex Female | 67 (1, 67-67, 67) | 59,006 (2.09%) | 3439 (0.12%) | 5672 (0.20%) | 35 (0, 20-44, 66) | 3989 (7.06%) | 1467 (2.93%) | 36 (0, 24 to 43, 65) | 1075 (17.43%) |
| Region Barcelona | 67 (1, 67-67, 67) | 82,702 (1.98%) | 7201 (0.17%) | 8694 (0.21%) | 36 (0, 19-44, 66) | 7604 (9.56%) | 2164 (3.10%) | 37 (0, 26-43, 65) | 2393 (19.20%) |
| Region Girona | 67 (1, 67-67, 67) | 9929 (1.78%) | 432 (0.078%) | 942 (0.17%) | 35 (0, 18-44, 66) | 657 (6.99%) | 217 (2.55%) | 32 (0, 17-41, 65) | 158 (17.48%) |
| Region Lleida | 67 (2, 67-67, 67) | 5621 (1.52%) | 211 (0.057%) | 728 (0.20%) | 33 (0, 16-43, 65) | 391 (7.35%) | 108 (2.45%) | 35 (0, 22-41, 64) | 94 (22.27%) |
| Region Tarragona | 67 (1, 67-67, 67) | 3748 (0.77%) | 196 (0.04%) | 958 (0.20%) | 30 (0, 16-42, 66) | 207 (5.82%) | 92 (3.00%) | 33 (0, 22-42, 64) | 45 (15.92%) |
| Charlson 0 | 67 (1, 67-67, 67) | 80,214 (1.76%) | 3627 (0.08%) | 1613 (0.035%) | 36 (0, 21-44, 66) | 5385 (7.01%) | 365 (0.55%) | 37 (0, 29-43, 65) | 583 (8.20%) |
| Charlson 1 | 67 (1, 67-67, 67) | 8996 (2.25%) | 1036 (0.26%) | 1607 (0.40%) | 33 (0, 15-43, 65) | 1031 (11.93%) | 519 (6.82%) | 36 (0, 23-43, 65) | 427 (24.22%) |
| Charlson 2 | 67 (2, 67-67, 67) | 6846 (1.91%) | 1409 (0.39%) | 2304 (0.64%) | 30 (0, 11-43, 65) | 1164 (17.70%) | 460 (8.16%) | 36 (0, 23-43, 65) | 559 (25.21%) |
| Charlson 3+ | 67 (2, 67-67, 67) | 5946 (2.20%) | 1969 (0.73%) | 5803 (2.15%) | 19 (0, 7-36, 64) | 1280 (22.53%) | 1237 (25.66%) | 31 (0, 17-41, 65) | 1123 (41.51%) |
| Autoimmune condition | 67 (2, 67-67, 67) | 6251 (2.29%) | 806 (0.29%) | 1285 (0.47%) | 34 (0, 16-44, 64) | 727 (12.10%) | 275 (5.09%) | 35 (0, 22-42, 65) | 326 (25.11%) |
| Chronic kidney disease | 67 (1, 67-67, 67) | 3934 (1.95%) | 1408 (0.70%) | 3710 (1.84%) | 19 (0, 7-35, 62) | 892 (23.67%) | 887 (27.36%) | 31 (0, 17-41, 64) | 819 (42.57%) |
| COPD | 67 (2, 67-67, 67) | 2359 (1.97%) | 740 (0.62%) | 1457 (1.22%) | 22 (0, 8-40, 65) | 463 (20.64%) | 261 (13.68%) | 33 (0, 19-42, 65) | 363 (36.26%) |
| Dementia | 67 (1, 67-67, 67) | 1933 (4.54%) | 429 (1.01%) | 2701 (6.35%) | 15 (0, 8-25, 59) | 294 (15.92%) | 990 (64.07%) | 21 (0.5, 12 to 34, 65) | 376 (64.59%) |
| Heart disease | 67 (1, 67-67, 67) | 10,732 (2.01%) | 2611 (0.49%) | 5686 (1.07%) | 28 (0, 10-41, 65) | 1903 (18.46%) | 1175 (13.20%) | 34 (0, 19-42, 65) | 1324 (34.22%) |
| Hyperlipidemia | 67 (1, 67-67, 67) | 10,820 (2.10%) | 1498 (0.29%) | 1580 (0.31%) | 34 (0, 15-43, 65) | 1507 (14.48%) | 381 (4.18%) | 37 (0, 27-43, 65) | 508 (20.40%) |
| Malignant neoplasm | 67 (1, 67-67, 67) | 5595 (1.92%) | 1327 (0.46%) | 3751 (1.29%) | 29 (0, 10-42, 65) | 991 (18.37%) | 588 (12.70%) | 34 (0, 20-43, 65) | 670 (34.19%) |
| Obesity | 67 (1, 67-67, 67) | 20,761 (2.24%) | 3216 (0.35%) | 3184 (0.34%) | 33 (0, 14-43, 66) | 3103 (15.52%) | 695 (3.94%) | 37 (0, 25-43, 65) | 1088 (20.03%) |
| Type 2 diabetes | 67 (2, 67-67, 67) | 6000 (1.89%) | 1644 (0.52%) | 2612 (0.82%) | 25 (0, 8-41, 64) | 1284 (22.30%) | 590 (11.77%) | 35 (0, 22-42, 65) | 710 (28.50%) |

Counts of less than 5 have been obscured to protect patient privacy.
COPD chronic obstructive pulmonary disease, IQR interquartile range.

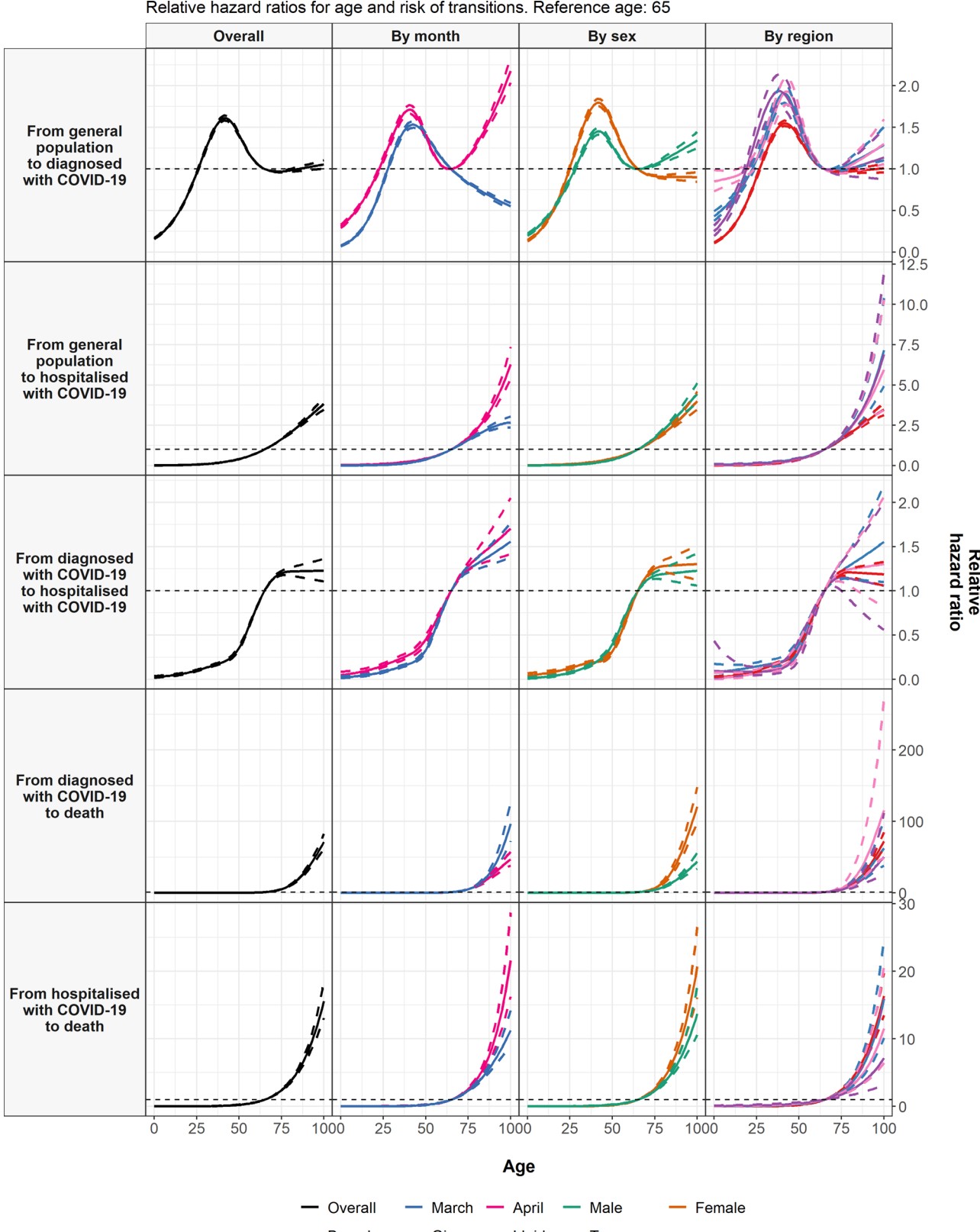

**Fig. 4 Age and COVID-19 transitions.** Estimated hazard ratios for age (relative to a reference age of 65) from cause-specific Cox models for each transition. Dotted lines represent 95% confidence intervals. The models shown included age as the only explanatory variable, with the overall models and models stratified by calendar month, sex, and region presented.

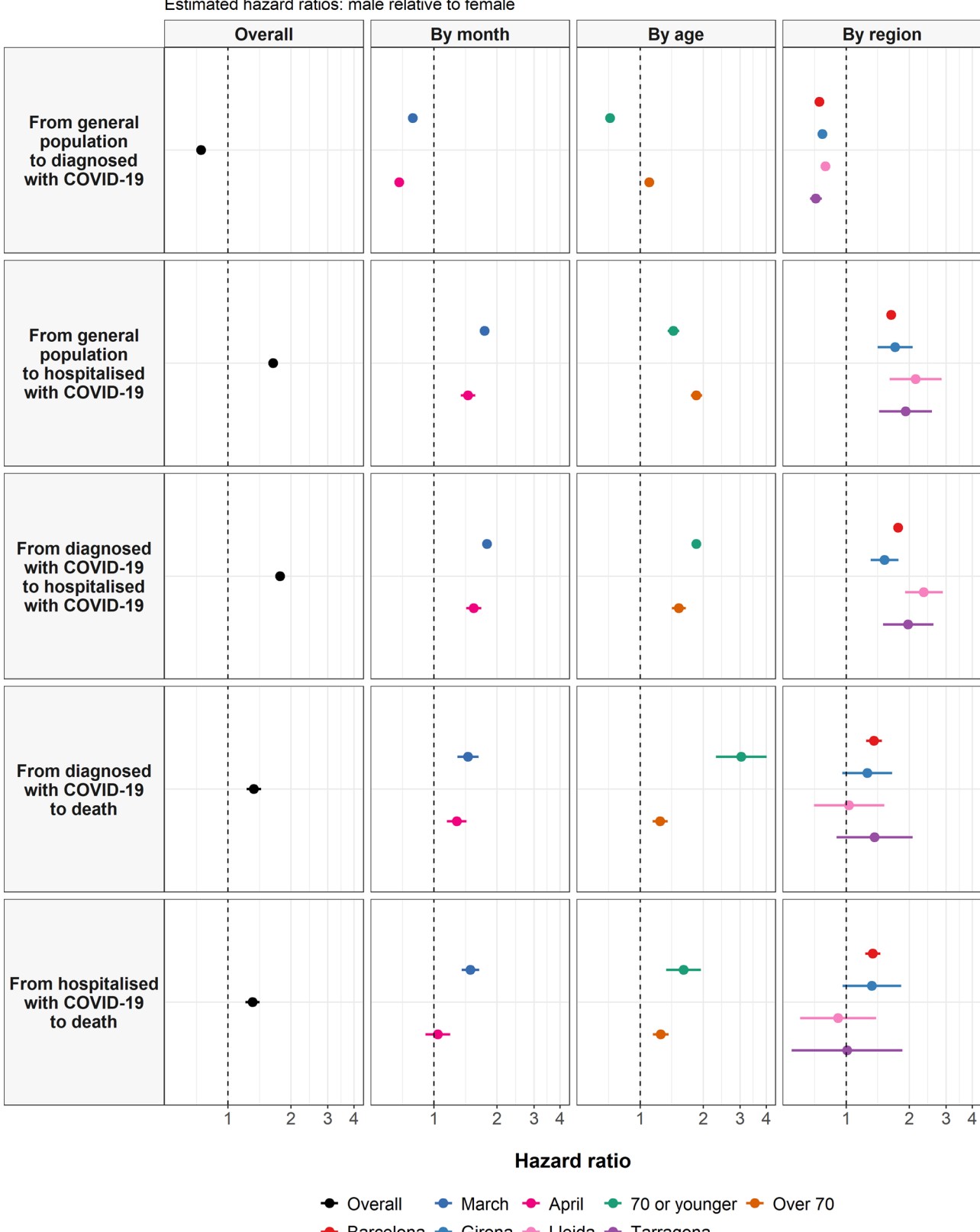

**Fig. 5 Sex and COVID-19 transitions.** Estimated hazard ratios for male sex (relative to female) from cause-specific Cox models for each transition. Points give estimated hazard ratios, with lines representing 95% confidence intervals. The models shown included sex and age as explanatory variables, with the overall models and models stratified by calendar month, sex, and region presented.

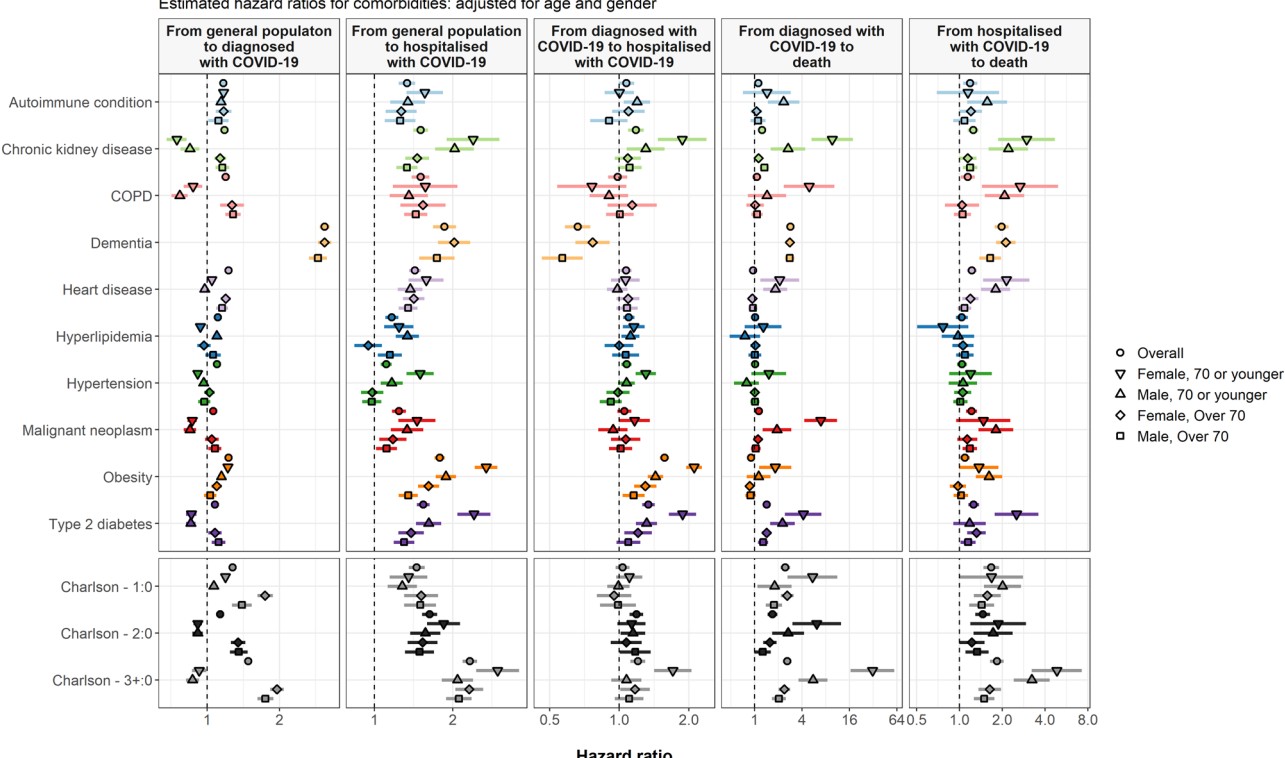

**Fig. 6 Comorbidities and COVID-19 transitions.** Estimated hazard ratios for comorbidities of interest from cause-specific Cox models for each transition. Points give estimated hazard ratios, with lines representing 95% confidence intervals. Models were estimated separately for each comorbidity of interest, with adjustment for age and sex. COPD chronic obstructive pulmonary disease.

first-wave of the epidemic in Spain. Over 100,000 outpatient COVID-19 diagnoses, close to 17,000 hospitalisations with COVID-19, and almost 5300 COVID-19 related deaths were observed between 1st March and 6th May. Of these deaths, half were among individuals who were diagnosed with COVID-19 in outpatient settings but were not seen to have been admitted to hospital.

Older age was consistently associated with increased risks of hospitalisation and mortality, most dramatically for risk of death following an outpatient diagnosis of COVID-19 without a subsequent hospital admission. There was though a notable peak in risk of outpatient diagnosis for people in middle-age during March, whereas in April risks were highest for oldest ages. A differential association was seen with sex; while women were at increased risk of outpatient diagnosis, men were consistently at higher risk of hospitalisation and death. Finally, the specific comorbidities studied (autoimmune condition, chronic kidney disease, chronic obstructive pulmonary disease, dementia, heart disease, hyperlipidemia, hypertension, malignant neoplasm, obesity, and type 2 diabetes), and a composite comorbidity index (Charlson Comorbidity Index) were associated with worse outcomes. In a number of cases the increase in risk associated with comorbidities appeared to be particularly pronounced for women aged under 70.

Older age has consistently been found to be a risk factor for worse outcomes in COVID-19. It has been associated with an increased risk of hospitalisation after testing positive for COVID-19[3,4], worse outcomes among those hospitalised[1,2,9], and an increased risk of COVID-19-related mortality among the general population[10]. Similarly, in our study we find older age to be associated with poorer outcomes. Our findings suggest that not only is older age associated with increased disease severity, but that there was also likely inequitable access to care

during the height of the pandemic in Spain. The increase in risk for age was most strongly seen for the transition from diagnosis to death with no admission to hospital seen between the two, while the risk of hospitalisation itself was also seen to fall at oldest ages. This may reflect, in part, rationing of health care resources, with younger patients likely prioritised for the receipt of hospital care, and intensive services if admitted. With half of the deaths in this study observed among those not hospitalised after diagnosis, who had an median age of 86, it is of utmost importance that similar individuals should be given appropriate access to care in the future.

Our findings on outpatient diagnoses of COVID-19 may provide further insights on community transmission of COVID-19. A peak in diagnosis of COVID-19 was seen among individuals around the age of 45 years old during March, but by April the risks of diagnosis were highest for those at oldest ages. While this could to some degree reflect differences in health seeking behaviour across age groups and changes in the way in which diagnoses were made and recorded, it likely also reflects differences in infection rates across age groups and transmission dynamics. A large study in Spain found that seroprevalence rose until plateauing at around age of 45 based on point-care-tests, but that seroprevalence was lower for those older than 85 compared to younger adults given immunoassay results[7]. Meanwhile, a seroprevalence study in Geneva found that those aged between 20 and 49 had the highest likelihood of being seropositive[11]. Consequently, while they face lower risks of poor outcomes if infected, younger age groups were infected and appear to have contributed to the spread of the virus in the community.

Being male has also been associated with worse prognosis in COVID-19, with increased risks for both hospitalisation among those tested[3], and worse outcomes among those hospitalised[2,3,9].

Our data are compatible with previous literature, with risks of hospital admission and mortality all increased for males. However, we found the opposite effect for outpatient diagnosis, with women being at increased risk of diagnosis with COVID-19 in the community. While this finding contrasts with two UK studies, which reported a higher risk of testing positive for SARS-CoV-2 among men[12,13], it is congruent with the findings from a study from China that reported a higher attack rate among women[14]. In Spain, seroprevalence was seen to be similar for males and females[7]. Further research is therefore needed to understand this finding, which could be explained by differences in exposure and/ or vulnerability to severe acute respiratory syndrome coronavirus 2 (SARS-CoV-2), and differences in health seeking behaviour between men and women.

Comorbidities have been associated with worse outcomes in COVID-19 in this study. In line with our findings, those with a higher Charlson score testing positive for COVID-19 have previously been seen to have an increased risk of hospitalisation and death[4]. Dementia was seen in our study to be associated with a much-increased risk of diagnosis with COVID-19. For those diagnosed, dementia was associated with a reduced risk of hospitalisation, but an increased risk of death without hospitalisation. As with age, the association between dementia and risk of diagnosis and hospitalisation with COVID-19 in the general population were seen to change over calendar time, with increased risks particularly pronounced in April. Other comorbidities were seen to be associated with worse outcomes, which is in accordance with previous findings on outcomes for individuals with chronic kidney disease, COPD, heart disease, hyperlipidemia, hypertension, cancer, obesity, and type 2 diabetes mellitus[2–4,12,13,15]. Our findings have also generally shown the associations between type 2 diabetes, obesity, and chronic kidney disease and COVID-19 outcomes to be most pronounced among women under 70. Fewer studies have assessed the relationship between autoimmune conditions and COVID-19 outcomes. As with other comorbidities, we found autoimmune conditions to be associated with an increased risk of poorer outcomes in COVID-19. Further research is though certainly merited to consider each of these conditions in turn and consider the impact of other factors such as these individuals' medication use.

Our study was informed by routinely-collected health care data with various interactions between individuals and the health system identified, covering outpatient diagnoses, COVID-19 testing, and hospitalisations, and with linked mortality data. This though unavoidably misses health outcomes experienced by individuals that do not lead to any interaction with the health care system, with both asymptomatic individuals and a sizeable proportion of mild symptomatic cases unlikely to be seen. Some of the clinical diagnoses observed in the study may also represent false positives, given the uncertainty surrounding the presentation of the disease during the study period. We did not require clinical diagnoses to be confirmed by the presence of a positive RT-PCR test, as such tests were not being routinely performed in outpatient settings in Spain during the study period. Deaths from COVID-19 where individuals were not tested or diagnosed beforehand will also not have been identified.

While providing a broad picture of clinical trajectories, each of the studied states and transitions can, and should, be considered in further detail. Analyses of the provision of intensive care admissions during hospitalisation is one such example, but would require further data with sufficient granularity on inpatient treatment. In addition, emergency room presentations that did not lead to a hospital admission were also not assessed in this study. Similarly, the set of comorbidities considered here are only a subset of the myriad set of conditions that are of interest when considering potential risk factors for COVID-19. Our classifications are broad, and we have not attempted to split conditions by severity or concurrent medication use.

The purpose of this study was descriptive in nature. The aim was not prediction, nor was it causal inference. In particular, it should be noted that associations between specific comorbidities and outcomes do not necessarily reflect a causal relationship. Assessing whether a particular chronic condition is the cause worse outcomes in COVID-19 will require further consideration of, and accounting for, relevant confounding factors. The associations which have been described here are also not immutable, but rather sit in the context of the first wave of COVID-19 in Spain. While there is substantial evidence that individuals at older age are at highest risk of severe disease if infected, effective shielding strategies would affect the risks of COVID-19 diagnoses and hospitalisations among this group. Similarly, ensuring that there is sufficient capacity to provide appropriate care would likely have a particular impact on the proportion of patients that die after a diagnosis with COVID-19 without a hospital admission.

In conclusion, in this descriptive study we provide a comprehensive overview of the natural history of symptomatic COVID-19 in Catalonia, Spain, during the first wave of the pandemic. The findings from this study can help inform the continued response to COVID-19 both in Spain and elsewhere. Our research has helped to reveal a clear need to protect at risk populations, particularly older people, whilst also considering middle-age populations at a particularly high risk of milder infections and therefore likely key in the community transmission of the disease.

## Methods

**Study design**. This study was informed by primary care records from Catalonia, Spain, which were linked to COVID-19 test results, hospital records, and mortality data. The resulting database was comprehensive, allowing for the identification of various key events in the progression of COVID-19 over the study period, which began on the 1st March 2020 and ended on the 6th May 2020. A multistate cohort model provided the framework for analysis, allowing for a systematic consideration of transitions for the general population to diagnosis of COVID-19, hospitalisation with COVID-19, and COVID-19 mortality.

**Study participants, setting, and data source**. Individual-level routinely-collected primary care data were extracted from the Information System for Research in Primary Care (SIDIAP; www.sidiap.org) database, which captures patient records from approximately 80% of the Catalan population and is representative in geography, age, and sex[16]. Linkage was made at an individual-level to COVID-19 RT-PCR testing data, hospital data, and regional mortality data.

The entire database has been mapped to the Observational Medical Outcomes Partnership (OMOP) Common Data Model (CDM). This provided a means of structuring the data to a standardised format, and allowed for the application of analytical tools developed by the open-science Observational Health Data Sciences and Informatics (OHDSI) network[17].

All individuals in SIDIAP as of the 1st March 2020 were identified. So that study participants had sufficient prior observation time for comorbidities to be identified, any individuals with less than one year of prior history available were excluded. As the index date for all individuals in the multistate model, described below, began on the 1st March, any individual who had a clinical diagnosis or positive test result for COVID-19 between the 1st January and 29th February 2020 was excluded. In addition, because the starting state in the model was representative of individuals living in the community in Catalonia, individuals who were hospitalised or a care home resident as of the 1st March 2020 were also excluded. Study follow-up began on the 1st March 2020 and ended on the 6th May 2020 (the last date of available data).

**Variables**. Individuals' age and sex were extracted. Health conditions, using individual's observed medical history were identified. The Charlson comorbidity index was calculated and scores were categorised as 0, 1, 2, or 3+. Ten specific health conditions were also extracted, all of interest as potential risk factors for the progression of COVID-19. Autoimmune condition (which included type 1 diabetes, rheumatoid arthritis, psoriasis, psoriatic arthritis, multiple sclerosis, systemic lupus erythematosus, Addison's disease, Grave's disease, Sjorgen's syndrome, Hashimoto thyroiditis, Myasthenia gravis, vasculitis, pernicious anaemia, celiac disease, scleroderma, sarcoidosis, ulcerative colitis, and Crohn's disease), chronic kidney disease, chronic obstructive pulmonary disease (COPD), dementia, heart disease, hyperlipidemia, hypertension, malignant neoplasm excluding non-

melanoma skin cancer, type 2 diabetes mellitus were each identified on the basis of diagnosis codes. Obesity was identified either by a diagnosis code, a record of a body mass index measurement between 30 and 60 kg/m$^2$, or a recorded weight between 120 and 200 kg within 5 years of the index date. The region in which an individual lived; Barcelona (inclusive of the Barcelona, Metro area north, Metro area south, and Catalunya Central health regions), Girona, Lleida (inclusive of both Lleida and Alt Pirineu i Aran health regions), or Tarragona (inclusive of the Tarragona and Terres de l'Ebre health regions) as of their index date was identified, as was whether they were a care home resident.

Study outcomes included an outpatient clinical diagnosis of COVID-19, a hospitalisation with COVID-19, and death. Outpatient COVID-19 diagnoses were identified on the basis of the first observation of a compatible clinical code being recorded for COVID-19 disease (such as ICD-10-CM codes B34.2 "Coronavirus infection, unspecified" and B97.29 "Other coronavirus as the cause of diseases classified elsewhere"). Hospitalisation with COVID-19 was identified as a hospital admission, a hospital stay of at least one night, where the individual had a positive RT-PCR test result or a clinical diagnosis of COVID-19 over the 21 days prior to their admission up to the end of their hospital stay[18]. Mortality was identified using region-wide mortality data, and so included both deaths during hospitalisations and in the community.

### Statistical methods

*Multistate model.* A multistate model provided the framework for the analysis. Multistate models allow for a consideration of individuals progression to multiple events of interest, extending on competing risk models by also describing transitions to intermediate events[19]. In the context of COVID-19, clinical diagnoses of the disease and hospitalisations with the disease can be considered as key intermediate events between not being infected (or at least, not having been identified as being so) on one end to death on the other.

The structure of the multi-state model used in this study is shown in Fig. 1. There are four states: general population, outpatient diagnosis with COVID-19, hospitalised with COVID-19, and death. The model is progressive with all individuals beginning in the general population state (as described above, individuals included in the study had no prior history of COVID-19 and were not hospitalised on their index date). Six transitions were possible: from general population to outpatient diagnosis with COVID-19, from general population to hospitalised with COVID-19 (i.e., individuals who did not get a clinical diagnosis prior to hospital admission), from general population to death, from outpatient diagnosis with COVID-19 to hospitalised with COVID-19, from outpatient diagnosis with COVID-19 to death, and from hospitalised with COVID-19 to death. Given the research objectives, the analyses focused on the five transitions directly related to COVID-19, with the general population to death used primarily as a means of accounting for the competing risk of death for the study population (although it should be noted that individuals who died with COVID-19, but had not been diagnosed or received a positive test will also be included in this transition).

The index date for all individuals, from which follow-up began, was 1st March 2020. For any given transition in the model, an individual's end of follow-up was whichever came first: their exit from the database (administrative censoring), the occurrence of the event of interest, the occurrence of a competing event, or the end of the study period (6th May 2020).

*Describing the association between age and risk of COVID-19 diagnosis, hospitalisation, and death.* Our objective here was to describe the overall association between age and outcomes seen during the first wave of COVID-19 in Catalonia. Consequently, the primary models of interest included age as the sole explanatory factor.

The relationship between age and the risk of each transition in the multistate model was assessed by estimating cause-specific Cox models, with hazard ratios and 95% confidence intervals (95% CIs) calculated. A non-linear relationship between age and the risk of transitions was considered by fitting age with a polynomial of degree 2 (i.e., quadratic) and with restricted cubic splines (with 3, 4, or 5 knots)[20]. Comparisons between these, and a model where age was assumed to have a linear relationship, were made using Akaike information criterion (AIC) and Bayesian Information Criterion (BIC). Where a more parsimonious model was seen to well-approximate a more complex one, the simpler model was favoured. Non-proportionality in hazards was considered through visual inspection of log–log plots.

To consider whether associations between age and outcomes varied by sex, models were also estimated separately for males and females. Similarly, models were also estimated separately for each region. To consider the impact of calendar time on the association between age and the risk of transitions from general population to outpatient diagnosis with COVID-19 and hospitalised with COVID-19, models were estimated separately for March and April (i.e., with follow-up split into these distinct periods). To consider the impact of calendar time on the association between age and subsequent transitions (i.e., from outpatient diagnosis with COVID-19 to hospitalised with COVID-19, outpatient diagnosis with COVID-19 to death, and from hospitalised with COVID-19 to death), models were estimated separately for those who arrived into these starting states in March and those who transitioned into them in April. Additionally, to consider the degree to

which observed associations could be explained by differences in comorbidity profiles across age groups, models were also estimated with additional adjustment subsequently made for comorbidities, with the Charlson score and the ten specific conditions of interest described above also included as explanatory factors in the model.

*Describing the association between sex and risk of COVID-19 diagnosis, hospitalisation, and death.* Our objective here was to describe the overall association between sex and outcomes, accounting for differences in age, during the first wave of COVID-19 in Catalonia. Therefore, the primary models included sex and age, with the sex the explanatory factor of interest.

As above, the relationship between sex and the risk of each transition in the multistate model was assessed by estimating cause-specific Cox models. Non-linearity in age was incorporated in the same way as for the previous research objective. To consider whether associations between sex and outcomes varied by age, models were also estimated separately for those aged 70 or younger and those aged over 70 (with age assumed to have a linear relationship within each of the strata). Stratified models were also estimated by region and calendar time. An additional set of models were also estimated with adjustment made for comorbidities.

*Describing the association between comorbidities and risk of transitions.* Our objective here was to describe the overall association between comorbidities and outcomes, accounting for differences in age and sex, during the first wave of COVID-19 in Catalonia. The primary models included the specific comorbidity of interest (the explanatory factor of interest), age, and sex.

The relationship between comorbidities and each transition was assessed by estimating cause-specific Cox models. Models were estimated for each comorbidity of interest separately, controlling for age and sex and with non-linearity in age incorporated in the same way as described above. To consider whether associations between comorbidities and outcomes varied by age and sex, models were subsequently estimated stratifying by both age (70 or younger or above 70) and sex, with age also included as an explanatory factor within each of the strata as a linear term. For dementia, models were only estimated for the older age strata. Stratified models were also estimated by region and calendar time.

**Reporting summary.** Further information on research design is available in the Nature Research Reporting Summary linked to this article.

### Data availability

In accordance with current European and national law, the data used in this study is only available for the researchers participating in this study. Thus, we are not allowed to distribute or make publicly available the data to other parties. However, researchers from public institutions can request data from SIDIAP if they comply with certain requirements. Further information is available online (https://www.sidiap.org/index.php/menu-solicitudes-en/application-proccedure) or by contacting Anna Moleras (amoleras@idiapjgol.org).

### Code availability

The mapping of source SIDIAP data to the OMOP CDM was facilitated by various open-source OHDSI software, which included usagi (https://github.com/OHDSI/Usagi), to help define mappings from source codes to the standard concepts used in the CDM, and Achilles (https://github.com/OHDSI/Achilles), to help assess data quality after mapping. Data analysis was performed in R version 4.0.0. The R packages used in the analysis included OHDSI CohortDiagnostics (https://github.com/OHDSI/CohortDiagnostics), numerous tidyverse packages[21], mstate[22], and rms[23]. The analytic code used in this study is freely available at https://github.com/SIDIAP/MultiStateCovid-19. Cohort definitions were adapted from those used in the OHDSI Seek COVER study and the OHDSI CHARYBDIS project[24]. A web application which summarises the cohort definitions used is available at https://livedataoxford.shinyapps.io/MultiStateCovidCohorts/.

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

## Acknowledgements

We would like to acknowledge the patients who suffered from or died of this devastating disease, and their families and carers. We would also like to thank the healthcare professionals involved in the management of COVID-19 during these challenging times, from primary care to intensive care units in the Catalan healthcare system. The analysis has made use of a range of open-source, free available tools provided by the OHDSI community. The phenotypes used in this study were developed, or adapted from, work performed during the OHDSI COVID-19 studyathon. The mapping of data to the OMOP CDM has been supported by a taskforce from the European Health Data and Evidence Network (EHDEN). This project is funded by the Health Department from the Generalitat de Catalunya with a grant for research projects on SARS-CoV-2 and COVID-19 disease organised by the Direcció General de Recerca i Innovació en Salut. This project has received support from the European Health Data and Evidence Network (EHDEN). EHDEN received funding from the Innovative Medicines Initiative 2 Joint Undertaking (JU) under grant agreement No 806968. The JU receives support from the European Union's Horizon 2020 research and innovation programme and EFPIA. The University of Oxford received a grant related to this work from the Bill & Melinda Gates Foundation (Investment ID INV-016201), and partial support from the UK National Institute for Health Research (NIHR) Oxford Biomedical Research Centre. D.P.A. is funded through a National Institute for Health Research (NIHR) Senior Research Fellowship (Grant number SRF-2018-11-ST2-004). The views expressed in this publication are those of the authors and not necessarily those of the NHS, the National Institute for Health Research or the Department of Health. A.P.-U. is supported by Fundacion Alfonso Martin Escudero and the Medical Research Council (grant numbers MR/K501256/1, MR/N013468/1).

## Author contributions

S.F.B., M.A., E.B., and T.D.S. mapped source data to the OMOP CDM and had accessed the data for the analysis. E.B., C.T., D.P.A., and T.D.S. led the data analysis. M.R. and E.R. performed a literature review. All authors were involved in the study conception and design, interpretation of the results, and the preparation of the manuscript.

## Competing interests

D.P.A. reports grants and other from AMGEN; grants, non-financial support and other from UCB Biopharma; grants from Les Laboratoires Servier, outside the submitted work; and Janssen, on behalf of IMI-funded EHDEN and EMIF consortiums, and Synapse Management Partners have supported training programmes organised by DPA's department and open for external participants. C.T. reports personal fees from Amgen, Boehringer ingelheim outside the submitted work. No other relationships or activities that could appear to have influenced the submitted work. All other authors declare no competing interests.
