## [Peer Review File · Nature Communications]

REVIEWERS' COMMENTS

Reviewer #2 (Remarks to the Author):

We thank the authors for addressing our comments. We are satisfied with the additional analyses and the revisions in the manuscript to emphasize the limitations of the findings they present. We hope our review was helpful and strengthened the manuscript.

We would like to thank the reviewers for the time and expertise invested in these reviews. Our responses to all comments are detailed below in bold (line numbers refer to those in the document with track changes).

We have made a number of substantive changes based on the comments received. In particular, we extracted information on the region of Catalonia in which study participants lived (Barcelona, Lleida, Girona, or Tarragona). It can be seen that while more COVID-19 outcomes were seen in the Barcelona region, the associations with age, sex, and comorbidities were broadly consistent across regions. We identified and excluded those individuals who were living in care homes. We also added a further analyses with respect to calendar time, and it can be seen that this did have a substantial effect on the association between age and the risk of an outpatient diagnosis with COVID-19, with those at oldest ages having a lower risk than a reference 65 year old in March, but an increased risk in April. Other results were seen to be consistent across time. Each of these are discussed below in more detail with respect to the relevant reviewer comments.

Yours sincerely,

Edward Burn (on behalf of all authors)

Reviewer #1 (Remarks to the Author):

The objective of this study was to assess the associations of age, sex and comorbidities with the risk of outpatient COVID-19 diagnosis, hospitalization and death, using health records in a cohort of over 100K patients from Catalonia, Spain. The main claims of the study are that: (1) half of deaths occurred outside of the hospital, mainly among elderly adults. (2) The relationship of age and risk of diagnosis had a double-ascent curve, with one peak in middle-age and increasing again in old age. (3) The risk of hospitalization and risk of death conditional on hospitalization increased with age. (4) In adjusted models, males had higher risk for all outcomes, except for outpatient COVID-19 diagnosis. (5) Number of comorbidities is associated with increased risk of outcomes. In particular, dementia was associated with large risk of diagnosis and death without hospitalization, and with low risk of hospitalization.

The main strengths of this work are: (1) its sample size, which allows for the generalizability of the results to populations similar to the underlying population; (2) the ability to follow patients starting with diagnosis in the outpatient setting; (3) the ability to capture outpatient mortality; and (4) the availability of comprehensive medical histories to properly assess comorbidities.

My feedback is categorized into Major and Minor issues below.

Major issues

1) MODELING. The main objectives of the study are to assess the relationship of age, sex and comorbidities to the outcomes. In such models, it is necessary to adjust for other covariates that are well known to be associated with the outcomes.

→ **We would like to first stress that the research questions we addressed in our analysis were descriptive in nature; summarising COVID-19 outcomes experienced during the first wave of COVID-19 in Catalonia and describing the associations between age, sex, and comorbidities and COVID-19 outcomes during this period. We have clarified our specific research questions at the end of the introduction:**

- **“In this study our first aim was to first summarise COVID-19 related outcomes in Catalonia as experienced during the first wave of the pandemic. Subsequently, we aimed to describe the associations between age, sex, and comorbidities and risks of COVID-19 diagnosis, hospitalisation with COVID-19, and a COVID-19-related death during the first wave of COVID-19 in Catalonia. It should be stressed that these latter research questions are inferential, assessing the existence of relationships but not the underlying mechanisms or reasons for them.”⁸ (lines 139-145)**

→ **Our questions were descriptive, not for prediction or causal inference. This distinction has important implications for the modelling strategy employed. Causal modelling, which should be done with much care to avoid potential pitfalls such as the ‘table 2 fallacy’, was beyond the scope of this study. Similarly, prediction which requires considerations of discrimination and calibration for example, was also beyond the scope of the study. In the text we have clarified this with:**

- **“The purpose of this study was descriptive in nature. The aim was not prediction, nor was it causal inference. In particular, it should be noted that associations between specific comorbidities and outcomes do not necessarily**

reflect a causal relationship. Assessing whether a particular chronic condition is the cause worse outcomes in COVID-19 will require further consideration of, and accounting for, relevant confounding factors.” (lines 510-512)

- ➔ For age, we have clarified that our objective was to describe the overall association between age and outcomes, and therefore our primary model was without adjustment for any other explanatory variables.
 - “Our objective here was to describe the overall association between age and outcomes seen during the first wave of COVID-19 in Catalonia. Consequently, the primary models of interest included age as the sole explanatory factor.” (lines 537 to 539)
- ➔ For sex, we have clarified that our objective was to describe the overall association between sex and outcomes, accounting for differences in age. Consequently, our primary model of interest was with age and sex included as explanatory factors.
 - “Our objective here was to describe the overall association between sex and outcomes, accounting for differences in age, during the first wave of COVID-19 in Catalonia. Therefore, the primary models included sex and age, with the sex the explanatory factor of interest.” (lines 569 to 571)
- ➔ For comorbidities, we have clarified that our specific objective was to describe the overall association accounting for differences in age and sex.
 - “Our objective here was to describe the overall association between comorbidities and outcomes, accounting for differences in age and sex, during the first wave of COVID-19 in Catalonia. The primary models included the specific comorbidity of interest (the explanatory factor of interest), age, and sex.” (lines 685 to 688)
- ➔ We believe there is still value in the formulation of these models with a descriptive aim, and that they provide broad, overall indications of associations between central patient characteristics and outcomes of interest as experienced during the first wave of COVID-19
- ➔ We have also noted that such associations, particularly from general population to being diagnosed or hospitalised, are not immutable.
 - “The associations which have been described here are also not immutable, but rather sit in the context of the first wave of COVID-19 in Spain. While there is substantial evidence that individuals at older age are at highest risk of severe disease if infected, effective shielding strategies would affect the risks of COVID-19 diagnoses and hospitalisations among this group. Similarly, ensuring that there is sufficient capacity to provide appropriate care would likely have a particular impact on the proportion of patients seen to have died after a diagnosis with COVID-19 but without a hospital admission.” (lines 529 to 536)

Adjusting for time is critical, since it is known that over time mortality rates and other outcomes have declined drastically in some regions. Adjusting for calendar time (i.e weeks since March 1, 2020) serves to adjust for changes in testing, treatments and clinical practice as well as the effects of outbreaks and lockdown measurements (see Tartof et al. 2020).

- ➔ We have added models stratified by calendar time. For the initial transitions from the *general population* state, this meant splitting observed follow-up into distinct windows for March and April. For subsequent transitions from the *outpatient diagnosis with COVID-19* and *hospitalised with COVID-19* states, this meant

estimating models separately for those who arrived into the state in March and those who arrived in April. We have added a description of this to the methods section

- **“To consider the impact of calendar time on the association between age and the risk of transitions from general population to outpatient diagnosis with COVID-19 and hospitalised with COVID-19, models were estimated separately for March and April (i.e. with follow up split into these distinct periods). To consider the impact of calendar time on the association between age and subsequent transitions (ie from outpatient diagnosis with COVID-19 to hospitalised with COVID-19, outpatient diagnosis with COVID-19 to death, and from hospitalised with COVID-19 to death), models were estimated separately for those who arrived into these starting states in March and those who transitioned into them in April.” (lines 652 to 653)**
- ➔ **The association between age and risk of outpatient diagnosis with COVID-19 did meaningfully change over time. Those at oldest ages were at a lower risk, relative to a 65-year-old, in March, but an increased risk in April. Other associations were broadly consistent over calendar time. We have added results from the models stratified by calendar time to Figures 4 and 5, and commented on them in the results and discussion sections of the paper.**

Similarly, other covariates such as at least one geographical variable or hospital-type variable would be ideal in the models. Examples of these variables are neighborhood median income, or population density, size or type of hospital (academic vs not-academic), number of beds, etc, depending on what is available to the authors (See Gupta et al 2020 for the association of number of ICU beds on mortality).

- ➔ **We extracted further information on which region individuals lived in (broadly defined as Barcelona, Girona, Lleida, or Tarragona), and have estimated models for all transitions stratified by region. The associations seen in the overall models were broadly consistent by region, and we have also added these results to Figures 4 and 5.**
- ➔ **Our analyses were descriptive, and it is beyond the scope of our study to undertake causal modelling (in which case, the adjustments suggested may well be made depending on the DAG guiding such modelling). We have tried to be clear with this throughout, for example**
 - **“In particular, it should be noted that associations between specific comorbidities and outcomes do not necessarily reflect a causal relationship. Assessing whether a particular chronic condition is the cause worse outcomes in COVID-19 will require further consideration of, and accounting for, relevant confounding factors.” (lines 527 to 529)**

2) INCLUSION CRITERIA. The methods section should be expanded to clarify how the following issues were addressed:

A) Data censoring. The study period was March 1 to May 6. How was the censoring of the data handled? For example, hospitalizations that were still ongoing at the end of the study. Please

clarify that if a hospitalization or death was not observed on May 6, the event time was right censored.

- ➔ **We have added further details on individual's time at risk for each of the transitions of interest, with right censoring used.**
 - **"The index date for all individuals, from which follow-up began, was the 1st March 2020. For any given transition in the model, an individual's end of follow-up was whichever came first: their exit from the database (administrative censoring), the occurrence of the event of interest, the occurrence of a competing event, or the end of the study period (6th May 2020)." (lines 629 to 633)**

B) Were hospitalizations assessed to ensure they were COVID-19 related or were all-cause hospitalizations considered? Is there a time window from COVID-19 diagnosis to hospitalization for it to be considered COVID-19 related? For example, if a patient was diagnosed on March 1st but hospitalized two months later on May 1st, is that hospitalization included as an outcome? Please clarify in the inclusion criteria.

- ➔ **The hospitalisations identified, used to identify entry into the *hospitalisation with COVID-19* state, were indeed only hospitalisations associated with COVID-19. To be considered a COVID-19-related hospitalisation, an individual was required to have positive PT-RT-PCR test result or a clinical diagnosis of COVID-19 over the 21 days prior to their admission up to the end of their hospital stay. This is the same definition as used in a recent paper, for which we have added a reference.**
 - **"Hospitalisation with COVID-19 was identified as a hospital admission, a hospital stay of at least one night, where the individual had a positive RT-PCR test result or a clinical diagnosis of COVID-19 over the 21 days prior to their admission up to the end of their hospital stay.¹⁹" (lines 602 to 605)**
- ➔ **In the example of a patient who was diagnosed on March 1st but hospitalized two months later on May 1st, this hospitalisation would not be considered as COVID-19 related and so it would not be included as an outcome (unless the individual had another diagnosis or a positive COVID-19 test result recorded between the 10th April and the end of their hospital stay).**

C) What is the index date in this study?

- ➔ **The index date for all individuals was the 1st March, and we have clarified this in the manuscript.**
 - **"The index date for all individuals, from which follow-up began, was the 1st March 2020." (line 629 to 630)**

How are multiple outpatient COVID-19 diagnoses handled or the case of multiple PCR tests?

- ➔ **An individual could only enter the *outpatient diagnosis with COVID-19* state once, with this transition identified on the basis of the first observation of a COVID-19 diagnosis after the index date. We have clarified this in the text.**
 - **"Outpatient COVID-19 diagnoses were identified on the basis of the first observation of a compatible clinical code" (lines 599 to 602)**

- All available diagnoses and PCR test results were used when identifying whether a hospitalisation was related to COVID-19, with a hospitalisation considered as related to COVID-19 if an individual had any COVID-19 diagnosis or positive test result observed over the 21 days prior to their admission up to the end of their hospital stay.
 - “Hospitalisation with COVID-19 was identified as a hospital admission where the individual had a positive RT-PCR test result or a clinical diagnosis of COVID-19 over the 21 days prior to their admission up to the end of their hospital stay.²⁰” (lines 602 to 605)

Based on the current description of eligibility (“no prior clinical diagnosis or positive test result for COVID-19”) it is not clear why in the results n=312 patients were excluded for “a prior clinical diagnosis of COVID-19”. Is it because this diagnosis occurred before March 1st? Please clarify.

- All study participants began in the *general population* state as of the 1st March, after which they were at risk of transitioning to subsequent outcome states. Therefore we excluded any individual who was seen to have a history of the outcomes were excluded (ie we excluded any individuals who had a COVID-19 diagnosis, positive test result, or COVID-19 hospitalisation prior to 1st March 2020).
- We have clarified this logic in the text
 - “As the index date for all individuals in the multi-state model, described below, began on the 1st March, any individual who had a clinical diagnosis or positive test result for COVID-19 between the 1st January and 29th February 2020 was excluded” (lines 579 to 573)

D) How is a hospitalization defined? Would single day visits to the ER be counted as a hospitalization? Is an overnight stay required? What about patients that visited the ER on repeated occasions following their COVID-19 diagnosis but didn’t stay overnight? Please clarify.

- Hospitalisations reflected inpatient admissions, defined by an overnight stay. We have clarified this in the text
 - “Hospitalisation with COVID-19 was identified as a hospital admission, a hospital stay of at least one night,” (lines 585 to 586)
- Emergency room presentations that did not lead to an admission were not included as an outcome. We have added this as a study limitation.
 - “In addition, emergency room presentations that did not lead to a hospital admission were also not assessed in this study.” (lines 602 to 605)

Clarifying these issues (A-D) in the inclusion/exclusion criteria description in the methods section would strengthen the conclusions of the study.

- We hope the above clarifications have helped to clarify the study design, with the exclusion criteria now explained as
 - “So that study participants had sufficient prior observation time for comorbidities to be identified, any individuals with less than one year of prior history available were excluded. As the index date for all individuals in

the multi-state model, described below, began on the 1st March, any individual who had a clinical diagnosis or positive test result for COVID-19 between the 1st January and 29th February 2020 was excluded. In addition, because the starting state in the model was representative of individuals living in the community in Catalonia, individuals who were hospitalised or a care home resident as of the 1st March 2020 were also excluded.” (lines 567 to 576)

→ We have also added an inclusion/ exclusion flow chart to the appendix (Supplementary Figure 1)

3) MISSINGNESS. Missingness in the covariates was not mentioned in the study. What are the rates of missingness in the variables in the model? And how was it handled? If there was missing information, incorporating multiple imputations is recommended.

- There was no missing data in the original set of variables extracted in the study. All individuals in SIDIAP have date of birth (used to calculate age) and their sex recorded. If someone doesn't have a record of a particular health condition recorded, they are assumed not to have that condition (i.e. if we see no records indicating that an individual has hypertension, the individual is considered as not having hypertension for the purposes of the study).
- Region did though have a small number of missing values (47 individuals out of the study population 5,586,521 did not have region recorded). These individuals were dropped when stratifying models by region, but contributed to all other analyses.

4) ASSUMPTIONS. Please include a description on how the model's assumptions were assessed, and if any assumption was violated, what measures were taken.

- The key assumption of the modelling approach used is that of proportionality in hazards over time. To assess this assumption we used visual inspection of log-log plots. We have clarified this in the methods
 - “Non-proportionality in hazards was considered through visual inspection of log-log plots.” (lines 648 to 649)
- The assumption of proportionality was seen to be violated in the case of age and the risk of the diagnosis with COVID-19, the consequences of which can be seen in the stratified models by calendar month (see Figure 4, with the corresponding log-log plot also presented in the appendix).

Minor issues

1) The clinical diagnosis of COVID-19 was assessed using ICD codes B34.2, and B97.29. However, U07.1 and U07.2 were not mentioned. These emergency COVID-19 codes were announced in March and became effective in April, which falls within the study period. (<https://www.who.int/classifications/icd/COVID-19-coding-icd10.pdf>) Please clarify whether these codes were considered in the COVID-19 diagnosis definition.

→ We have checked and neither of these codes have been recorded in our data.

1) Results section. The age transitions paragraph mentions both median and average age. However, based on Table 1, it appears all these values are medians. Please harmonize the way age is reported.

→ **We have clarified this, and reported this as the median throughout.**

1) The study would benefit from additional references that include large scale health-records based studies. For example: Bello-Chavolla et al. (2020), Tartof et al. (2020), Williamson et al (2020), Gupta et al. (2020).

→ **We have added reference to Tartof et al. (2020) and Gupta et al. (2020) which we think are particularly relevant in contextualising our findings.**

2) The reference section needs to be revised. References do not have a consistent formatting and key information is missing, such as the journal name or source (e.g. References #5, #12, among others).

→ **We have checked and amended our references in line with the formatting for Nature Communications.**

Final note

Based on the objectives of the study and the nature of the data, it appears that the most adequate word to use throughout the paper is “sex” instead of “gender”. Otherwise, please clarify.

→ **We have changed gender to sex throughout the manuscript.**

References:

Bello-Chavolla et al. (2020) Predicting mortality due to SARS-CoV-2: A mechanistic score relating obesity and diabetes to COVID-19 outcomes in Mexico. *J Clin Endocrinol Metab.* 2020;105:dga346.

Gupta et al. (2020) Factors associated with death in critically ill patients with coronavirus disease 2019 in the US. *JAMA Intern Me.* doi:10.1001/jamainternmed.2020.3596

Tartof et al. (2020) Obesity and mortality among patients diagnosed with COVID-19: Results from an integrated health care organization. *Ann Intern Med.* 2020;M20-3742. doi:10.7326/M20-3742

Williamson et al. (2020) Factors associated with COVID-19-related death using OpenSAFELY.

Nature 584, 430–436.

Reviewer #2 (Remarks to the Author):

Synopsis:

Burn et al present a 10-week longitudinal analysis of the natural progression of COVID-19 in Catalonia (Spain). Using comprehensive data capturing approximately 80% of the national Catalanian population, the authors cross-link community database information with hospital data and mortality data, to explore three primary endpoints: COVID-19 diagnosis, hospitalisation due to COVID-19, or death.

Present findings included 109,367 outpatient COVID-19 diagnoses, 18,091 hospitalisations and 5,585 deaths; with males and those with medical comorbidities (including dementia) associated with higher likelihood of COVID-19 diagnosis, hospitalisation and death; concluded by the authors as findings important for informing health policy in Catalonia.

Strengths:

The manuscript offers important insights into the natural progression of COVID 19 in Catalonia, Spain. Spain is an area afflicted by particularly tragic COVID-19 spread and outcomes; the first European country to record over half a million cases and with recent rates of new infection more than twice that of the next worst-affected country in Europe (France). [1,2] An understanding of the natural progression of the disease and of at-risk populations, in this region of Europe in particular, is of great importance.

The authors should be commended on the the size of their dataset and how comprehensively they have linked data from the community and the hospital with national mortality data. This important opportunity to describe disease progression longitudinally, capturing such a broad scope of the national population, is to the authors' credit.

Major issues:

The first issue pertains to the selection bias introduced by who got tested for COVID-19 in the Catalanian region during the study period. In an ideal situation, everyone would have been tested.

→ **In our study we did not establish an outpatient COVID-19 test positive state as community testing during the first wave of COVID-19 in Catalonia in March and April 2020 was indeed not done in a systematic manner. In March and April PCR tests were primarily reserved for patients with severe disease who were, or about to be,**

hospitalised and, as time progressed, were prioritised for specific at-risk populations, such as care home residents.

- The sole COVID-19 outpatient state in our model is that of a clinical diagnosis of COVID-19, as reported in our primary care database. These individuals were not required to have their diagnosis confirmed by a positive test result. This was because of the practical reality that few such patients received a test at the start of the pandemic in Catalonia.
- We did, however, use COVID-19 testing data for the confirmation that a hospitalisation was related to COVID-19.
- We believe this use of testing data minimises the risks of selection bias into the study. To note, we start from the general population, who could transition to a hospitalisation without being identified as having COVID-19 in the outpatient setting.
- There are of course, however, unavoidable limitations created due to the way in which individuals were identified (or, more pertinently, not identified) as having COVID-19 pandemic.
 - “Our study is informed by routinely-collected health care data with various interactions between individuals and the health system identified, covering outpatient diagnoses, COVID-19 testing, and hospitalisations, and with linked mortality data. This though unavoidably misses health outcomes experienced by individuals that do not lead to any interaction with the health care system, with both asymptomatic individuals and a sizeable proportion of mild symptomatic cases unlikely to be seen. Some of the clinical diagnoses observed in the study may also represent false positives, given the uncertainty surrounding the presentation of the disease during the study period. We did not require clinical diagnoses to be confirmed by the presence of a positive RT-PCR test, as such tests were not being routinely performed in outpatient settings in Spain during the study period. Deaths from COVID-19 where individuals were not tested or diagnosed beforehand will also not have been identified.” (lines 504 to 515)

While it is unclear whether most endpoints reached in the present study were attributed toward nursing home populations, collateral findings suggest they might be (i.e. the average age of death of non-hospitalised people was 87, and those with dementia had a substantially higher hazard ratio of diagnosis or death (both consistent with previous findings from a nursing home population); and the known higher rates of surveillance and testing during this period in Spain, as described by authors). [3] The COVID-19 nursing home crisis is well documented – accounting for over 40% of all COVID-19 deaths in the US [4,5] and upwards of 70% of all COVID-19 deaths in some European countries like Sweden. Nursing homes are uniquely poorly equipped to stop virus spread; lacking resources for virus containment (including personal protective equipment), limited in their ability to effectively socially distance, and with staff who are often under-paid and under-trained. [3,6] The importance of nursing home funding, regulation and support in containing virus spread has been discussed elsewhere [6] and is of ongoing importance in regions where nursing home residents are particularly affected (as may be the case here).

Such significant differences between nursing homes vs the general community make stratification by place of residence (i.e. nursing home vs. community) important in

ameliorating concerns that the risk ratios are being confounded by unobservable variables in the current specifications of the model. Is it dementia that is correlated with COVID-19 infection and death, or is it residency in a nursing home?

Given the frequency of surveillance/testing in Spanish nursing homes over the study period (as stated by authors), the COVID-19 status for those in nursing homes would presumably be more well-known vs for those in the community. This introduces the risk of information (sampling) bias if not properly controlled for. At the time of this study, in Spain, around 35% of infections in the general community were asymptomatic and only 10% were diagnosed. [2,7]

Comparatively, there were substantially higher rates of asymptomatic infection in nursing homes (up to 57%), [3] where (despite this) there was more comprehensive surveillance/testing (given the known outbreaks); with subsequently well-captured diagnoses (irrespective of symptomatology) in nursing homes vs the general community. Having such relative oversampling from one population is important to control for, and indeed should be done here through either stratified analyses or controlling for residence in the multivariable model.

The risks factors associated with primary endpoints (male gender, co-morbidities, dementia) may be heavily influenced by a predominately-nursing home population that was sampled, and very different risk factors may exist for a population in the general community.

- **We entirely agree that the care home residents are a distinct population, for whom transmission dynamics, disease surveillance, and access to care differed to that of the general community.**
- **Given that the focus of our study is on individuals living in the community as of the 1st March 2020, we have identified those individuals who were care home residents as of this date and excluded them from the study.**
- **A thorough analysis of the outcomes of those living in care homes is merited, but is beyond the scope of this study (as it will require additional information, such as identifying which individuals lived in which care home which is not available in our data).**

More geographical/regional information (even in the form of a covariate for “county” or “region” in multivariable analysis) are important confounders to adjust for as they affect the spread of COVID-19 in the community. Catalonia’s heterogeneous population with regard to age, socioeconomic status and access to healthcare (particularly urban vs rural areas) is known; differences further compounded by substantial geographical variation between regions. [8] Subsequently, the natural history of COVID-19 and risk factors predicting progression are likely to vary by region. Implementation of COVID-19 research findings to date has been limited by the fact that data is usually restricted to a single geographical region, [9] rather than population-wide, national data (as is used here). Findings are consequently not necessarily generalizable to areas with fundamental differences in typology: demographics, socioeconomic factors and even political leaning. The size and scope of the dataset used here represents an opportunity to explore risk of disease progression by county typology, which may elucidate additional drivers in community spread (e.g. age as a reflection of mobility) that have been previously identified with integrated datasets. [10]

- **As described above in response to a related comment from Reviewer 1, we extracted further information on which region individuals lived in (broadly defined as Barcelona, Girona, Lleida, or Tarragona), and have estimated models for all**

transitions stratified by region. The associations seen in the overall models were broadly consistent by region, and we have also added these results to Figures 4 and 5.

- Further, more granular examination of geography and population density and COVID-19 spread and outcomes is certainly an interesting topic, but beyond the scope of this paper.

In addition, adjusting for testing rate and positivity rate will address concerns that certain subsets of the population are being confounded by unobservable covariates. For example, the paper posits a double peaked age risk profile of diagnosis but does not distinguish between two distinct phenomena that could lead to the observed risk profile. First, the underlying population infection status by age is indeed double peaked, or second, the testing is not being carried at random in the observed sample and leads to the observed peaks. To clarify this, it would be useful to break out the positivity rates by age group, by county, and even by time period, as explained below. And similar to controlling for nursing home residency to disentangle the relationship between dementia and COVID-19 infection and death, the region-age-week testing rate and region-age-week positivity rate should be included in the Cox regression to account for information bias.

- To clarify, in our study, outpatient diagnoses were not tied to testing, but were rather based on clinical diagnoses. As such the trends in testing during the period seem to be of less relevance. Moreover, given that our objectives are descriptive (which we have tried to clarify above), confounding is not of a concept of direct relevance to the study (as we are explicitly not addressing questions of causal inference).
- Observed clinical diagnoses do not necessarily reflect the infection status in the community, with individuals presenting to primary care with symptomatic COVID-19. However, we do think they can be instructive, particularly in the context of the ENE-COVID seroprevalence study (Pollán et al. Prevalence of SARS-CoV-2 in Spain (ENE-COVID): a nationwide, population-based seroepidemiological study. Lancet)
- We have tried to be balanced in our summary of these implications in the text:
 - “Our findings on outpatient diagnoses of COVID-19 may provide further insights on community transmission of COVID-19. A peak in diagnosis of COVID-19 was seen among individuals around the age of 45 years old during March, but by April the risks of diagnosis were highest for those at oldest ages. While this may to some degree reflect differences in health seeking behaviour across age groups and changes in the way in which diagnoses were made and recorded, it likely also reflects differences in infection rates across age groups and transmission dynamics. A large and well-conducted study in Spain found that seroprevalence rose until plateauing at around age of 45 based on point-care-tests, but that seroprevalence was lower for those older than 85 compared to younger adults given immunoassay results.”⁷ (lines 454 to 463)

Time is a crucial covariate that should be adjusted for. Despite the relatively short 10-week period the data was collected over, the incidence of COVID-19 diagnoses and death, as influenced by policy implementations, changed significantly over time. For example, in just a

14-day period leading up to August, Spain was one of only 6 European countries to report a >30% increase in the incidence of infection, [1] recording up to 260 infections per 100,000 (over twice the rate of the next most affected European country (France)); [2] incidence rates as high as the disease peak in March. By stratifying time into five 2-week blocks (or similar) +/- adjusting for time in the multivariable analysis, relevant effect modification of risk factors by time periods (with corresponding restrictions to borders and travel, lockdown, return to work etc.) can be modeled. [11]

- **As described above in response to a similar comment from Reviewer 1, we have added models stratified by calendar time as suggested. For the initial transitions from the *general population* state, this meant splitting observed follow-up into distinct windows for March and April. For subsequent transitions from the *outpatient diagnosis with COVID-19* and *hospitalised with COVID-19* states, this meant estimating models separately for those who arrived into the state in March and April. We have added a description of this to the methods section**
 - **“To consider the impact of calendar time on the association between age and the risk of transitions from general population to outpatient diagnosis with COVID-19 and hospitalised with COVID-19, models were estimated separately for March and April (i.e. with follow up split into these distinct periods). To consider the impact of calendar time on the association between age and subsequent transitions (ie from outpatient diagnosis with COVID-19 to hospitalised with COVID-19, outpatient diagnosis with COVID-19 to death, and from hospitalised with COVID-19 to death), models were estimated separately for those who arrived into these starting states in March and those who transitioned into them in April.” (lines 652 to 660)**
- **The association between age and risk of outpatient diagnosis with COVID-19 did meaningfully change over time. Those at oldest ages were at a lower risk, relative to a 65-year-old, in March, but an increased risk. Other associations were broadly consistent over calendar time. We have added results from the models stratified by calendar time to Figures 4 and 5, and commented on them in the results and discussion sections of the paper.**

There are many other important socioeconomic factors that were not accounted for in the present study: namely race or ethnicity, poverty, and education. These could have been obtained from the Spanish census data by county or postal code of the patient. Racial disparities in COVID-19 diagnosis, hospitalisation and death are well described, however are likely due to the complex interplay between healthcare access and literacy, co-morbidities, occupation and other socioeconomic factors rather than by race alone. [12-15] The exploration of the non-biologic determinants of health and COVID-19, is important in truly understanding vulnerability to infection and death.

- **We entirely agree that many of these are important factors in COVID-19 and that any causal modelling should account for them wherever possible. However, such information is not currently available in our data, and it was beyond the scope of our descriptive study to assess them.**
- **In our study we have summarised the broad associations between age, sex, and comorbidities and COVID-19 during the first wave in Catalonia. We have tried to**

clarify that what we have not done, and is beyond the scope of this single study, was disentangle the reasons for the observed associations.

- **“In this study our first aim was to first summarise COVID-19 related outcomes in Catalonia as experienced during the first wave of the pandemic. Subsequently, we aimed to describe the associations between age, sex, and comorbidities and risks of COVID-19 diagnosis, hospitalisation with COVID-19, and a COVID-19-related death during the first wave of COVID-19 in Catalonia. It should be stressed that these latter research questions are inferential, assessing the existence of relationships but not the underlying mechanisms or reasons for them.⁸” (lines 139 to 145)**
- **“The purpose of this study was descriptive in nature. The aim was not prediction, nor was it causal inference. In particular, it should be noted that associations between specific comorbidities and outcomes do not necessarily reflect a causal relationship. Assessing whether a particular chronic condition is the cause worse outcomes in COVID-19 will require further consideration of, and accounting for, relevant confounding factors.” (lines 525 to 529)**

Minor issue:

Further exploration of missing data is warranted. Presumably some data was missing (i.e. 20% of the population were not included in the dataset). Exploration of who was disproportionately affected by missingness (by age, gender, race etc) should be outlined, so as to highlight which populations may be underrepresented and underreported.

- ➔ **There was no missing data in the original set of variables extracted in the study.**
- ➔ **Region did though have a small number of missing values (47 individuals out of the study population 5,586,521 did not have region recorded). These individuals were dropped when stratifying models by region, but contributed to all other analyses.**
- ➔ **We would not consider the 20% of the Catalan population not captured to be missing data in the typical sense (which we consider to be with regards to missing values for variables for individuals included in a database). The population of SIDIAP has previously been found to be representative of the population of Catalonia as a whole, and so we believe our results to be generalisable to this population.**
 - **“Individual-level routinely-collected primary care data were extracted from the Information System for Research in Primary Care (SIDIAP; www.sidiap.org) database, which captures patient records from approximately 80% of the Catalan population and is representative in geography, age, and sex.¹⁶” (lines 560 to 562)**

References

1. Coronavirus disease 2019 (covid-19) in the eu/eea and the uk – eleventh update: Resurgence of cases. Stockholm: European Centre for Disease Prevention and Control;2020.
2. Dombey D. Covid: Why spain is hit worse than the rest of europe. 2020. Accessed September, 2020.
3. Stern S, Klein D. Stockholm city’s elderly care and covid19: Interview with barbro karlsson. Nature Public Health Emergency Collection. 2020(19):1-12. doi: 10.1007/s12115-020-00508-0.
4. More than 40% of u.S. Coronavirus deaths are linked to nursing homes. 2020; <https://www.nytimes.com/interactive/2020/us/coronavirus-nursing-homes.html>. Accessed September, 2020.
5. K Yourish KRL, D Ivory, M Smith. One-third of all u.S. Coronavirus deaths are nursing home residents or workers. 2020; <https://www.nytimes.com/interactive/2020/05/09/us/coronavirus-cases-nursing-homes-us.html>. Accessed September, 2020.
6. Werner RM, Hoffman AK, Coe NB. Long-term care policy after covid-19 — solving the nursing home crisis. New England Journal of Medicine. 2020;383(10):903-905. doi: 10.1056/nejmp2014811.
7. Pollán M, Pérez-Gómez B, Pastor-Barriuso R, et al. Prevalence of sars-cov-2 in spain (ene-covid): A nationwide, population-based seroepidemiological study. The Lancet. 2020;396(10250):535-544. doi: 10.1016/s0140-6736(20)31483-5.
8. Catalonia, spain. World Health Organization: World Health Organization;2018.
9. Coronavirus (covid-19) data in the united states. Coronavirus (Covid-19) Data in the United States 2020; <https://github.com/nytimes/covid-19-data>. Accessed September, 2020.
10. Lai Y, Charpignon M, Ebner D, Celi L. Unsupervised learning for county-level typological classification for covid-19 research. Intelligence-Based Medicine. 2020;1-2:1-6. doi: <https://doi.org/10.1016/j.ibmed.2020.100002>.
11. Prem K, Liu Y, Russell T, Kucharski A, Eggo R, Davies N. The effect of control strategies to reduce social mixing on outcomes of the covid-19 epidemic in wuhan, china: A modelling study. The Lancet Public Health. 2020;5(5):E261-E270.
12. Chowkwanyun M, Reed A. Racial health disparities and covid-19 — caution and context. N Engl J Med. 2020;383(3):201-203.
13. Patel J. Poverty, inequality and covid-19: The forgotten vulnerable. Elsevier Public Health Emergency Collection. 2020;183:110-111.
14. Adhikari S, Pantaleo NP, Feldman JM, Ogedegbe O, Thorpe L, Troxel AB. Assessment of community-level disparities in coronavirus disease 2019 (covid-19) infections and deaths in large us metropolitan areas. JAMA Network Open. 2020;3(7):e2016938. doi: 10.1001/jamanetworkopen.2020.16938.
15. Health equity considerations and racial and ethnic minority groups. Centers for Disease Control and Prevention: Centers for Disease Control and Prevention;2020.

Leo Anthony Celi
Edward Christopher Dee
William Mitchell

REVIEWER COMMENTS

Reviewer #1 (Remarks to the Author):

The objective of this study was to assess the associations of age, sex and comorbidities with the risk of outpatient COVID-19 diagnosis, hospitalization and death, using health records in a cohort of over 100K patients from Catalonia, Spain. The main claims of the study are that: (1) half of deaths occurred outside of the hospital, mainly among elderly adults. (2) The relationship of age and risk of diagnosis had a double-ascent curve, with one peak in middle-age and increasing again in old age. (3) The risk of hospitalization and risk of death conditional on hospitalization increased with age. (4) In adjusted models, males had higher risk for all outcomes, except for outpatient COVID-19 diagnosis. (5) Number of comorbidities is associated with increased risk of outcomes. In particular, dementia was associated with large risk of diagnosis and death without hospitalization, and with low risk of hospitalization.

The main strengths of this work are: (1) its sample size, which allows for the generalizability of the results to populations similar to the underlying population; (2) the ability to follow patients starting with diagnosis in the outpatient setting; (3) the ability to capture outpatient mortality; and (4) the availability of comprehensive medical histories to properly assess comorbidities.

My feedback is categorized into Major and Minor issues below.

Major issues

1) MODELING. The main objectives of the study are to assess the relationship of age, sex and comorbidities to the outcomes. In such models, it is necessary to adjust for other covariates that are well known to be associated with the outcomes. Adjusting for time is critical, since it is known that over time mortality rates and other outcomes have declined drastically in some regions. Adjusting for calendar time (i.e weeks since March 1, 2020) serves to adjust for changes in testing, treatments and clinical practice as well as the effects of outbreaks and lockdown measurements (see Tartof et al. 2020). Similarly, other covariates such as at least one geographical variable or hospital-type variable would be ideal in the models. Examples of these variables are neighborhood median income, or population density, size or type of hospital (academic vs not-academic), number of beds, etc, depending on what is available to the authors (See Gupta et al 2020 for the association of number of ICU beds on mortality).

2) INCLUSION CRITERIA. The methods section should be expanded to clarify how the following issues were addressed:

A) Data censoring. The study period was March 1 to May 6. How was the censoring of the data handled? For example, hospitalizations that were still ongoing at the end of the study. Please clarify that if a hospitalization or death was not observed on May 6, the event time was right censored.

B) Were hospitalizations assessed to ensure they were COVID-19 related or were all-cause hospitalizations considered? Is there a time window from COVID-19 diagnosis to hospitalization for it to be considered COVID-19 related? For example, if a patient was diagnosed on March 1st but hospitalized two months later on May 1st, is that hospitalization included as an outcome? Please clarify in the inclusion criteria.

C) What is the index date in this study? How are multiple outpatient COVID-19 diagnoses handled or the case of multiple PCR tests? Based on the current description of eligibility ("no prior clinical diagnosis or positive test result for COVID-19") it is not clear why in the results n=312 patients were excluded for "a prior clinical diagnosis of COVID-19". Is it because this diagnosis occurred before March 1st? Please clarify.

D) How is a hospitalization defined? Would single day visits to the ER be counted as a hospitalization? Is an overnight stay required? What about patients that visited the ER on repeated occasions following their COVID-19 diagnosis but didn't stay overnight? Please clarify.

Clarifying these issues (A-D) in the inclusion/exclusion criteria description in the methods section would strengthen the conclusions of the study.

3) MISSINGNESS. Missingness in the covariates was not mentioned in the study. What are the rates of missingness in the variables in the model? And how was it handled? If there was missing information, incorporating multiple imputations is recommended.

4) ASSUMPTIONS. Please include a description on how the model's assumptions were assessed, and if any assumption was violated, what measures were taken.

Minor issues

1) The clinical diagnosis of COVID-19 was assessed using ICD codes B34.2, and B97.29. However, U07.1 and U07.2 were not mentioned. These emergency COVID-19 codes were announced in March and became effective in April, which falls within the study period. (<https://www.who.int/classifications/icd/COVID-19-coding-icd10.pdf>) Please clarify whether these codes were considered in the COVID-19 diagnosis definition.

1) Results section. The age transitions paragraph mentions both median and average age. However, based on Table 1, it appears all these values are medians. Please harmonize the way age is reported.

1) The study would benefit from additional references that include large scale health-records based studies. For example: Bello-Chavolla et al. (2020), Tartof et al. (2020), Williamson et al (2020), Gupta et al. (2020).

2) The reference section needs to be revised. References do not have a consistent formatting and key information is missing, such as the journal name or source (e.g. References #5, #12, among others).

Final note

Based on the objectives of the study and the nature of the data, it appears that the most adequate word to use throughout the paper is "sex" instead of "gender". Otherwise, please clarify.

References:

Bello-Chavolla et al. (2020) Predicting mortality due to SARS-CoV-2: A mechanistic score relating obesity and diabetes to COVID-19 outcomes in Mexico. *J Clin Endocrinol Metab.* 2020;105:dga346.

Gupta et al. (2020) Factors associated with death in critically ill patients with coronavirus disease 2019 in the US. *JAMA Intern Me.* doi:10.1001/jamainternmed.2020.3596

Tartof et al. (2020) Obesity and mortality among patients diagnosed with COVID-19: Results from an integrated health care organization. *Ann Intern Med.* 2020;M20-3742. doi:10.7326/M20-3742

Williamson et al. (2020) Factors associated with COVID-19-related death using OpenSAFELY. *Nature* 584, 430–436.

Reviewer #2 (Remarks to the Author):

Synopsis:

Burn et al present a 10-week longitudinal analysis of the natural progression of COVID-19 in Catalonia (Spain). Using comprehensive data capturing approximately 80% of the national Catalan population, the authors cross-link community database information with hospital data and mortality data, to explore three primary endpoints: COVID-19 diagnosis, hospitalisation due to COVID-19, or death.

Present findings included 109,367 outpatient COVID-19 diagnoses, 18,091 hospitalisations and 5,585 deaths; with males and those with medical comorbidities (including dementia) associated with higher likelihood of COVID-19 diagnosis, hospitalisation and death; concluded by the authors as findings important for informing health policy in Catalonia.

Strengths:

The manuscript offers important insights into the natural progression of COVID 19 in Catalonia, Spain. Spain is an area afflicted by particularly tragic COVID-19 spread and outcomes; the first European country to record over half a million cases and with recent rates of new infection more than twice that of the next worst-affected country in Europe (France). [1,2] An understanding of the natural progression of the disease and of at-risk populations, in this region of Europe in particular, is of great importance.

The authors should be commended on the the size of their dataset and how comprehensively they have linked data from the community and the hospital with national mortality data. This important opportunity to describe disease progression longitudinally, capturing such a broad scope of the national population, is to the authors' credit.

Major issues:

The first issue pertains to the selection bias introduced by who got tested for COVID-19 in the Catalan region during the study period. In an ideal situation, everyone would have been tested. While it is unclear whether most endpoints reached in the present study were attributed toward nursing home populations, collateral findings suggest they might be (i.e. the average age of death of non-hospitalised people was 87, and those with dementia had a substantially higher hazard ratio of diagnosis or death (both consistent with previous findings from a nursing home population); and the known higher rates of surveillance and testing during this period in Spain, as described by authors). [3] The COVID-19 nursing home crisis is well documented – accounting for over 40% of all COVID-19 deaths in the US [4,5] and upwards of 70% of all COVID-19 deaths in some European countries like Sweden. Nursing homes are uniquely poorly equipped to stop virus spread; lacking resources for virus containment (including personal protective equipment), limited in their ability to effectively socially distance, and with staff who are often under-paid and under-trained. [3,6] The importance of nursing home funding, regulation and support in containing virus spread has been discussed elsewhere [6] and is of ongoing importance in regions where nursing home residents are particularly affected (as may be the case here).

Such significant differences between nursing homes vs the general community make stratification by place of residence (i.e. nursing home vs. community) important in ameliorating concerns that the risk ratios are being confounded by unobservable variables in the current specifications of the model. Is it dementia that is correlated with COVID-19 infection and death, or is it residency in a nursing home?

Given the frequency of surveillance/testing in Spanish nursing homes over the study period (as stated by authors), the COVID-19 status for those in nursing homes would presumably be more well-known vs for those in the community. This introduces the risk of information (sampling) bias if not properly controlled for. At the time of this study, in Spain, around 35% of infections in the general community were asymptomatic and only 10% were diagnosed. [2,7] Comparatively, there were substantially higher rates of asymptomatic infection in nursing homes (up to 57%), [3] where (despite this) there was more comprehensive surveillance/testing (given the known outbreaks); with subsequently well-captured diagnoses (irrespective of symptomatology) in nursing homes vs the general community. Having such relative oversampling from one population is important to control for, and indeed should be done here through either stratified analyses or controlling for residence in the multivariable model. The risk factors associated with primary endpoints (male gender, co-morbidities, dementia) may be heavily influenced by a predominately-nursing home population that was sampled, and very different risk factors may exist for a population in the general community.

More geographical/regional information (even in the form of a covariate for "county" or "region" in multivariable analysis) are important confounders to adjust for as they affect the spread of COVID-19 in the community. Catalonia's heterogeneous population with regard to age, socioeconomic status and access to healthcare (particularly urban vs rural areas) is known; differences further compounded by substantial geographical variation between regions. [8] Subsequently, the natural history of COVID-19 and risk factors predicting progression are likely to vary by region. Implementation of COVID-19 research findings to date has been limited by the fact that data is usually restricted to a single geographical region, [9] rather than population-wide, national data (as is used here). Findings are consequently not necessarily generalizable to areas with fundamental differences in typology: demographics, socioeconomic factors and even political leaning. The size and scope of the dataset used here represents an opportunity to explore risk of disease progression by county typology, which may elucidate additional drivers in community spread (e.g. age as a reflection of mobility) that have been previously identified with integrated datasets. [10]

In addition, adjusting for testing rate and positivity rate will address concerns that certain subsets of the population are being confounded by unobservable covariates. For example, the paper posits a double peaked age risk profile of diagnosis but does not distinguish between two distinct phenomena that could lead to the observed risk profile. First, the underlying population infection status by age is indeed double peaked, or second, the testing is not being carried out at random in the observed sample and leads to the observed peaks. To clarify this, it would be useful to break out the positivity rates by age group, by county, and even by time period, as explained below. And similar to controlling for nursing home residency to disentangle the relationship between dementia and COVID-19 infection and death, the region-age-week testing rate and region-age-week positivity rate should be included in the Cox regression to account for information bias.

Time is a crucial covariate that should be adjusted for. Despite the relatively short 10-week period the data was collected over, the incidence of COVID-19 diagnoses and death, as influenced by policy implementations, changed significantly over time. For example, in just a 14-day period leading up to August, Spain was one of only 6 European countries to report a >30% increase in the incidence of infection, [1] recording up to 260 infections per 100,000 (over twice the rate of the next most affected European country (France)); [2] incidence rates as high as the disease peak in March. By stratifying time into five 2-week blocks (or similar) +/- adjusting for time in the multivariable analysis, relevant effect modification of risk factors by time periods (with corresponding restrictions to borders and travel, lockdown, return to work etc.) can be modeled. [11]

There are many other important socioeconomic factors that were not accounted for in the present study: namely race or ethnicity, poverty, and education. These could have been obtained from the Spanish census data by county or postal code of the patient. Racial disparities in COVID-19

diagnosis, hospitalisation and death are well described, however are likely due to the complex interplay between healthcare access and literacy, co-morbidities, occupation and other socioeconomic factors rather than by race alone. [12-15] The exploration of the non-biologic determinants of health and COVID-19, is important in truly understanding vulnerability to infection and death.

Minor issue:

Further exploration of missing data is warranted. Presumably some data was missing (i.e. 20% of the population were not included in the dataset). Exploration of who was disproportionately affected by missingness (by age, gender, race etc) should be outlined, so as to highlight which populations may be underrepresented and underreported.

References

1. Coronavirus disease 2019 (covid-19) in the eu/eea and the uk – eleventh update: Resurgence of cases. Stockholm: European Centre for Disease Prevention and Control;2020.
2. Dombey D. Covid: Why spain is hit worse than the rest of europe. 2020. Accessed September, 2020.
3. Stern S, Klein D. Stockholm city's elderly care and covid19: Interview with barbro karlsson. Nature Public Health Emergency Collection. 2020(19):1-12. doi: 10.1007/s12115-020-00508-0.
4. More than 40% of u.S. Coronavirus deaths are linked to nursing homes. 2020; <https://www.nytimes.com/interactive/2020/us/coronavirus-nursing-homes.html>. Accessed September, 2020.
5. K Yourish KRL, D Ivory, M Smith. One-third of all u.S. Coronavirus deaths are nursing home residents or workers. 2020; <https://www.nytimes.com/interactive/2020/05/09/us/coronavirus-cases-nursing-homes-us.html>. Accessed September, 2020.
6. Werner RM, Hoffman AK, Coe NB. Long-term care policy after covid-19 — solving the nursing home crisis. New England Journal of Medicine. 2020;383(10):903-905. doi: 10.1056/nejmp2014811.
7. Pollán M, Pérez-Gómez B, Pastor-Barriuso R, et al. Prevalence of sars-cov-2 in spain (ene-covid): A nationwide, population-based seroepidemiological study. The Lancet. 2020;396(10250):535-544. doi: 10.1016/s0140-6736(20)31483-5.
8. Catalonia, spain. World Health Organization: World Health Organization;2018.
9. Coronavirus (covid-19) data in the united states. Coronavirus (Covid-19) Data in the United States 2020; <https://github.com/nytimes/covid-19-data>. Accessed September, 2020.
10. Lai Y, Chappignon M, Ebner D, Celi L. Unsupervised learning for county-level typological classification for covid-19 research. Intelligence-Based Medicine. 2020;1-2:1-6. doi: <https://doi.org/10.1016/j.ibmed.2020.100002>.
11. Prem K, Liu Y, Russell T, Kucharski A, Eggo R, Davies N. The effect of control strategies to reduce social mixing on outcomes of the covid-19 epidemic in wuhan, china: A modelling study. The Lancet Public Health. 2020;5(5):E261-E270.
12. Chowkwanyun M, Reed A. Racial health disparities and covid-19 — caution and context. N Engl J Med. 2020;383(3):201-203.
13. Patel J. Poverty, inequality and covid-19: The forgotten vulnerable. Elsevier Public Health Emergency Collection. 2020;183:110-111.
14. Adhikari S, Pantaleo NP, Feldman JM, Ogedegbe O, Thorpe L, Troxel AB. Assessment of community-level disparities in coronavirus disease 2019 (covid-19) infections and deaths in large us metropolitan areas. JAMA Network Open. 2020;3(7):e2016938. doi: 10.1001/jamanetworkopen.2020.16938.

15. Health equity considerations and racial and ethnic minority groups. Centers for Disease Control and Prevention: Centers for Disease Control and Prevention;2020.

Leo Anthony Celi
Edward Christopher Dee
William Mitchell